# Logit-KL Flow Matching: Non-Autoregressive Text Generation via Sampling-Hybrid Inference

**Egor Sevriugov**
LigandPro, Moscow, Russia
Applied AI institute, Moscow, Russia
`egor.sevriugov@gmail.com`

**Nikita Dragunov**
FusionBrain Lab, AXXX, Moscow, Russia

**Anton Razzhigaev**
FusionBrain Lab, AXXX, Moscow, Russia

**Andrey Kuznetsov**
FusionBrain Lab, AXXX, Moscow, Russia

**Ivan Oseledets**
AXXX, Moscow, Russia
Applied AI institute, Moscow, Russia

## Abstract

Non-autoregressive (NAR) language models offer notable efficiency in text generation by circumventing the sequential bottleneck of autoregressive decoding. However, accurately modeling dependencies in discrete sequences remains challenging in this paradigm. In this work, we advance the field of NAR generation by applying conditional flow matching (CFM) methods grounded in geometrically principled interpolation, specifically leveraging Kullback-Leibler (KL) divergence geodesics, which correspond to linear interpolation in logit space. We rigorously establish that maximizing conditional likelihood in this setting precisely recovers the flow matching velocity field, supplying the theoretical justification for this approach in sequence modeling. To address practical performance gaps of *basic* inference, we propose a novel empirical *sampling* strategy that iteratively denoises and re-noises, along with a *hybrid* scheme that integrates our *sampling* method with *basic* procedure. Across unconditional and conditional text and code infilling, the approach improves perplexity and downstream metrics over prior NAR baselines under matched settings.

## 1 Introduction

Non-autoregressive (NAR) language models have emerged as efficient alternatives to traditional autoregressive models in NLP by generating all tokens simultaneously. However, capturing complex dependencies in discrete textual data remains challenging without sequential modeling.

We investigate conditional flow matching (CFM) methods for text generation, building on recent advances such as Discrete Flow Matching (DFM) Gat et al. (2024), Dirichlet Flow Matching Stärk et al. (2024), and Fisher-Flow Davis et al. (2024), which represent tokens as one-hot vectors in a $V - 1$-dimensional simplex. These methods interpolate a sequence of distributions $\rho_t$ from an initial $\rho_0$ to a data distribution $\rho_1$; for text, the latter is sampled as discrete sequences in the simplex. Prior work identifies issues with naive linear interpolation in simplex space Stärk et al. (2024). We propose instead using KL-geodesics, equivalent to linear interpolation in logit space, to better capture the underlying geometry.

Our CFM framework leverages this interpolation, training with a denoiser maximizing the conditional likelihood $p_\theta(x_1 \mid x_t)$, enabling tractable approximation of the joint distribution. While theoretical guarantees previously existed only for single-token predictions, we show that maximiz-

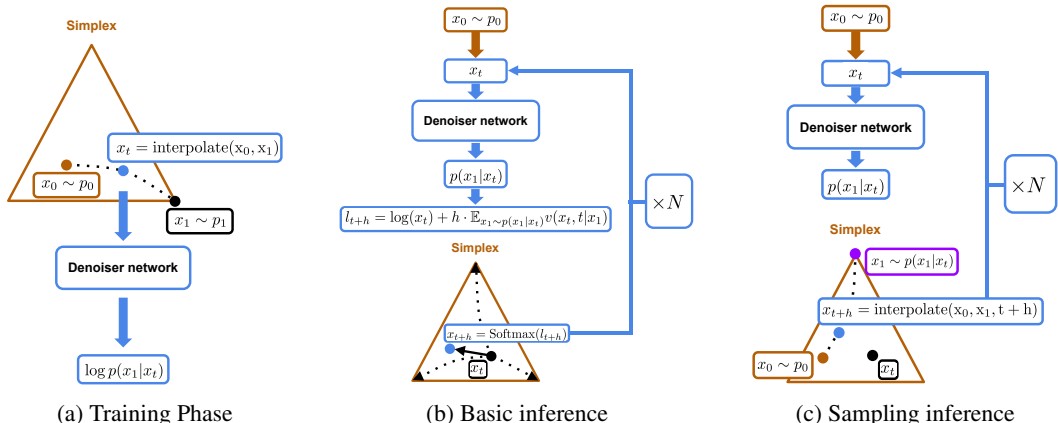

(a) Training Phase        (b) Basic inference        (c) Sampling inference

Figure 1: **Overview of the Proposed Approach:** *Training*: Sample $x_0 \sim p_0$ (uniform distribution on simplex), $x_1 \sim p_1$ (target distribution represented by samples); interpolate to obtain $x_t$. The denoiser network predicts $\log p_\theta(x_1|x_t)$, trained via log-probability maximization. *Inference*: For *basic* inference, numerically solve an ODE with vector field: $\mathbb{E}_{x_1 \sim p_\theta(x_1|x_t)}[v(x_t, t \mid x_1)]$ using Euler method with $N$ steps and a step size of $h = 1/N$. Alternatively, in *sampling* inference, interpolate between $x_0 \sim p_0$ and $x_1 \sim p(x_1|x_t)$ at each step.

ing this conditional likelihood still exactly recovers the flow matching velocity field in logit space for sequence modeling, lending theoretical support to our approach.

Standard inference procedures with this framework yield suboptimal results, so we introduce a novel sampling strategy: given a state $x_t$, we sample $x_1$ from $p(x_1 \mid x_t)$ and re-noise it to $x_{t+h}$, iterating this process. Despite the lack of full theoretical analysis, this method yields stronger empirical results. We further propose a hybrid inference scheme blending our basic and sampling strategies, yielding improved performance on tasks such as text generation, conditional question answering, and code infilling (see Figure 1).

Our contributions are:

- Using KL-geodesic (logit-space linear) interpolation for flow matching in discrete sequences.
- Theoretical analysis showing conditional likelihood maximization exactly recovers the flow matching velocity field for logit-space interpolation.
- A novel sampling and hybrid inference strategy with strong empirical results.
- Empirical improvements: at least 27% lower perplexity for unconditional generation (Fine-FineWeb), and at least 17%, 26% BLEU boosts for conditional tasks (Lamini Instruction, WMT 14 de-en); plus 56% and 14% gains in Pass@1 and Pass@10 for code infilling where 10% of the code lines were omitted. Prior methods are trained and evaluated under the same setup to ensure a fair comparison.

## 2 BACKGROUND

Flow matching Lipman et al. (2023) constructs a deterministic transport from a simple *base* distribution $\rho_0$ (e.g., $\mathcal{N}(0, I)$) to an unknown *data* distribution $\rho_1$ given by samples. It introduces a time-dependent density $\rho(x, t)$ and velocity field $v(x, t)$ for $t \in [0, 1]$ that satisfy the mass-conservation (Liouville) equation

$$\partial_t \rho(x, t) = -\nabla \cdot \big(\rho(x, t)\, v(x, t)\big), \tag{1}$$

with boundary conditions $\rho(\cdot, 0) = \rho_0$ and $\rho(\cdot, 1) = \rho_1$. Once $v$ is known, samples are generated by integrating the characteristic ODE

$$\frac{dx_t}{dt} = v(x_t, t), \qquad x_{t=0} \sim \rho_0, \tag{2}$$

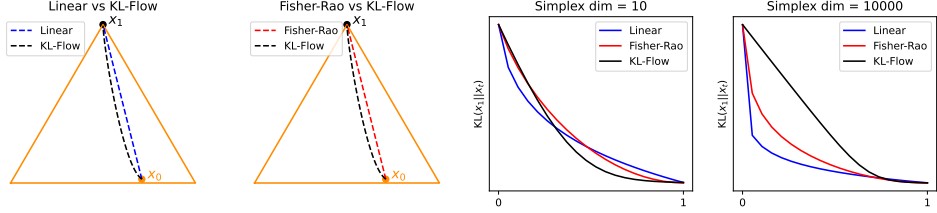

Figure 2: Qualitative and quantitative comparison of three distinct classes of geodesics on the probability simplex: Linear, Fisher–Rao, and KL–Flow. The first two panels juxtapose the trajectories of the Linear and Fisher–Rao interpolations against the KL–Flow interpolation. The rightmost panels depict the temporal evolution of the divergence $\mathrm{KL}(x_1 \| x_t)$ for two different simplex dimensions, $|\mathcal{V}| = 10$ and $|\mathcal{V}| = 10\,000$.

Table 1: Perplexity (lower is better) obtained by 150M–parameter language models trained on the *FineFineWeb* corpus under the Linear, Fisher–Rao, KL-Flow geodesics.

| Geodesic | Llama 2 | GPT 3 | GPT 2 |
|---|---|---|---|
| Linear | 1344 | 15418 | 13881 |
| Fisher–Rao | 192 | 298 | 379 |
| KL-Flow | 41 | 53 | 62 |

and taking $x_{t=1}$ as a draw from $\rho_1$.

**Learning the velocity field (conditional flow matching).** Because $v$ is unknown, it is approximated with a neural network $v_\theta(x, t)$ using interpolation between initial and target data samples. Let $\gamma_t(x_0, x_1)$ be any interpolation with $\gamma_0 = x_0$ and $\gamma_1 = x_1$; draw $t \sim \mathcal{U}[0, 1]$, $x_0 \sim \rho_0$, and $x_1 \sim \rho_1$, and set $x_t = \gamma_t(x_0, x_1)$. Define the vector field $v(x_t, t \,|\, x_0, x_1) \coloneqq \frac{d}{dt}\gamma_t(x_0, x_1)$. The conditional flow matching objective is

$$\mathcal{L}_{\mathrm{CFM}}(\theta) \;=\; \mathbb{E}_{t, x_0, x_1}\Big[\| v_\theta(\gamma_t(x_0, x_1), t) - v(\gamma_t(x_0, x_1), t \,|\, x_0, x_1)\|_2^2\Big]. \tag{3}$$

At inference, we integrate the ODE with $v_\theta$ from $t=0$ to $t=1$ to obtain samples.

## 3 CONDITIONAL FLOW MATCHING FOR DISCRETE SEQUENCES

In language modelling, the terminal random variable $x_1$ is a one–hot vector (a vertex of the $(V-1)$–simplex). Following Stärk et al. (2024), we take the initial distribution $\rho_0$ to be the uniform (Dirichlet($\mathbf{1}$)) measure on the simplex, so $x_0 \sim \rho_0$ is a strictly positive probabilistic token mixture. A central design choice in flow matching is the *interpolation* between $x_0$ and $x_1$.

**KL geodesic on the simplex.** While linear interpolation in probability space is possible, its drawbacks for discrete data have been documented (Stärk et al., 2024); Fisher–Rao geodesics have also been proposed (Davis et al., 2024). We instead use the geodesic induced by the Kullback–Leibler (KL) divergence—the canonical information–theoretic discrepancy on the simplex.

**Definition 3.1** (KL geodesic). For $t \in [0, 1]$, the *KL–geodesic* joining $x_0$ and $x_1$ is

$$x_t \;=\; \frac{x_0^{1-t} x_1^t}{\sum_{i=1}^{V} x_{0,i}^{1-t} x_{1,i}^t} \;\equiv\; C_t \, x_0^{1-t} x_1^t, \tag{4}$$

where $C_t$ normalizes $x_t$ onto the simplex.

It is *linear in logits*, $l_t = (1 - t)\log x_0 + t \log x_1$ with $x_t = \mathrm{Softmax}(l_t)$. Moreover, KL–geodesics preserve a *usable learning signal*: as shown in Fig. 2, $\mathrm{KL}(x_1 \| x_t)$ decays substantially more slowly along the KL path—especially for large vocabularies ($|\mathcal{V}|{=}10{,}000$)—whereas Linear and Fisher–Rao paths collapse $\mathrm{KL}(x_1 \| x_t)$ near zero too early (for $t$ close to 0), effectively turning the

transport into a one–shot step and depriving the model of informative gradients over most of the time horizon. Empirically, Table 1 shows that training with Linear or Fisher–Rao objectives yields markedly worse perplexity, consistent with this geometric analysis.

**Logit parameterization**   Write $l_0 = \log x_0$ and $l_1 = \log x_1$, and define the logit–linear interpolation $l_t = (1 - t)l_0 + t\, l_1$, so that $x_t = \mathrm{Softmax}(l_t)$. Because $\log$ is undefined at zero, we use a standard $\beta$–smoothed target for the one–hot $x_1$,

$$x_1 \;=\; (1 - \beta)\, \delta_i + \frac{\beta}{V}\, \mathbf{1}, \qquad \beta \in (0, 1),$$

where $\delta_i$ is the canonical basis vector of the realized token and $\mathbf{1}$ is the all–ones vector. Equivalently, in logit space we could write linear ODE:

$$\frac{dl_t}{dt} \;=\; l_1 - l_0, \tag{5}$$

so the KL path is a straight line in logits whose image under $\mathrm{Softmax}$ remains intrinsic to the simplex.

## 3.1   DENOISING OBJECTIVE

**Single–token case.**   Consider the special case in which the sequence length equals one. The entire input is then represented by a single vector whose dimensionality matches the vocabulary size. As introduced in Definition 3.1 the KL-geodesic reproduces the conditional flow–matching objective equation 3:

$$\mathcal{L}_{\mathrm{CFM}}(\theta) = \mathbb{E}_{t, x_0, x_1} \left\| v_\theta(x_t, t) - (l_1 - l_0) \right\|^2, \tag{6}$$

where $x_t = \mathrm{Softmax}(l_t)$ denotes the intermediate point obtained by applying the softmax map to the logit vector $l_t = (1 - t)\, l_0 + t\, l_1$. The quantities $l_0$ and $l_1$ are, respectively, the logits generating the initial state $x_0$ and the target state $x_1$ after projection onto the probability simplex. Both the conditional vector field $v(x_t, t \mid x_0, x_1) = l_1 - l_0$ and its learnable counterpart $v_\theta(x_t, t)$ admit the following reparametrisation in terms of $l_t$:

$$l_1 - l_0 = \frac{l_1 - l_t}{1 - t}, \qquad v_\theta(x_t, t) = \frac{\hat{v}_\theta(x_t, t) - l_t}{1 - t}, \tag{7}$$

Substituting the identities in equation 7 into the loss equation 6 transforms the original objective into a denoising-style regression problem in which the model must recover the clean target logit $l_1 = \log x_1$ from the corrupted observation $x_t$:

$$\mathcal{L}_{\mathrm{CFM}}(\theta) \;=\; \mathbb{E}_{t, x_0, x_1} \left\| \hat{v}_\theta(x_t, t) - l_1 \right\|^2. \tag{8}$$

**Proposition 3.2.** *Let $\mathcal{L}_{\mathrm{CFM}}(\theta)$ be defined as in equation 8. For every $t \in (0, 1)$ and every $x_t$ the function*

$$\hat{v}_\theta^\star(x_t, t) \;=\; \mathbb{E}_{x_1 \sim p(x_1 \mid x_t)}\, l_1 \tag{9}$$

*is the (almost surely) unique minimiser of the loss equation 8.*

**Corollary 3.3.** *Suppose we approximate the true conditional $p(x_1 \mid x_t)$ with a parametric model $p_\theta(x_1 \mid x_t)$. Then an estimate of the vector field compatible with equation 7 is*

$$v(x_t, t) = \frac{1}{1 - t} \left( \mathbb{E}_{x_1 \sim p_\theta(x_1 \mid x_t)}\, l_1 - l_t \right). \tag{10}$$

*The subscript $\theta$ is omitted in $v(x_t, t)$ to emphasise that learning proceeds through the conditional density $p_\theta(x_1 \mid x_t)$, rather than through direct parametrisation of the vector field itself.*

**Sequences of length $S$**   We now extend the analysis from the single-token setting to sequences that contain exactly $S$ tokens. As a prior over sequences we assume $S$ independent Dirichlet distributions, each defined on the $(V - 1)$–simplex associated with the vocabulary of size $V$. In contrast, the "clean" or target distribution $p_1$ is supported on the vertices of the Cartesian product of simplices. Following the prescriptions in Stärk et al. (2024); Gat et al. (2024), we interpolate each token independently along the KL–geodesic. Consequently, the logit representation becomes an $S \times V$ matrix $l_t$ whose $k$-th row $l_t^{(k)}$ corresponds to token $k$.

Fixing an index $k \in \{1, \ldots, S\}$ and specialising Equation equation 9 to the present context yields

$$\hat{v}_\theta^{(k)}(x_t, t) = \mathbb{E}_{x_1 \sim p(x_1 | x_t)} \, l_1^{(k)}, \tag{11}$$

where $l_1^{(k)}$ denotes the logits that would generate the clean token $x_1^{(k)}$.

**Proposition 3.4.** *For the KL–geodesic described above, the expression in equation 11 factorises over individual tokens, and the optimal vector field for the $k$-th coordinate can be written as*

$$\hat{v}_\theta^{(k)}(x_t, t) = \mathbb{E}_{x_1^{(k)} \sim p(x_1^{(k)} | x_t)} \, l_1^{(k)}, \tag{12}$$

*where $p(x_1^{(k)} \mid x_t)$ is the marginal conditional distribution associated with the $k$-th token.*

Consequently, under the KL–geodesic, computing the optimal velocity field reduces to evaluating the exact marginal posteriors $p(x_1^{(k)} \mid x_t)$ for each token $k$ independently. In practice we approximate these posteriors with a parametric model $p_\theta(x_1^{(k)} \mid x_t)$. We draw $x_1 \sim p_1$ (from the data distribution) and $t \sim \mathcal{U}(0, 1)$, set $x_0 \sim p_0$, and form $x_t = \text{Softmax}\big((1 - t) \log x_0 + t \log x_1\big)$. The model outputs token-wise conditionals $p_\theta(x_1^{(k)} \mid x_t)$, for which we minimize the sequence-level NLL:

$$\mathcal{L} = -\mathbb{E}_{t, x_1 \sim p(x_1), x_t \sim p(x_t | x_1)} \sum_{k=1}^{S} \log p_\theta\big(x_1^{(k)} \mid x_t\big), \tag{13}$$

A practical realisation of the conditional model $p_\theta(x_1^{(k)} \mid x_t)$ can be obtained by adapting a Transformer architecture: the standard causal attention is replaced with bidirectional attention so that the representation of each token has access to the entire sequence $x_t$, and an additional conditioning mechanism is introduced to incorporate the continuous time variable $t$.

## 4 INFERENCE: ITERATIVE SAMPLING SCHEME

We present three complementary inference procedures under the KL–geodesic interpolation introduced earlier: a *deterministic KL–flow integrator*, a *stochastic iterative sampler*, and a *hybrid* routine that combines both. Unless stated otherwise, logits evolve along the logit–linear path

$$l_t = (1 - t) \, l_0 + t \, l_1, \qquad x_t = \text{Softmax}(l_t).$$

### 4.1 DETERMINISTIC INFERENCE VIA KL–FLOW

Within classical flow matching, samples are generated by numerically integrating the ODE associated with the KL–geodesic. For the interpolation in Definition 3.1, the logit vector obeys the linear ODE

$$\frac{dl_t}{dt} = \frac{l_1 - l_t}{1 - t}. \tag{14}$$

Algorithm 1 implements an explicit scheme (Euler with step size $h = 1/N$) that advances $t$ from 0 to 1. In experiments we refer to this baseline as **KL–flow (basic)**.

### 4.2 STOCHASTIC INFERENCE BY DIRECT SIMULATION

An alternative is to *simulate* the one–step transport induced by a small time increment $h > 0$. Conditioning on the current iterate $x_t$, the next iterate admits the Markov factorization

$$p(x_{t+h} | x_t) = \int p(x_{t+h} | x_1) \, p(x_1 | x_t) \, dx_1. \tag{15}$$

The exact posterior $p(x_1 \mid x_t)$ is intractable at the sequence level. The optimization of objective from equation 13 gives the product of tokenwise marginals produced by the denoiser:

$$p_\theta(x_1 | x_t) = \prod_{k=1}^{S} p_\theta\big(x_1^{(k)} | x_t\big).$$

Table 2: Summary of inference methods.

| Method | Description | Update rule | Limitations |
|---|---|---|---|
| KL-Flow (basic) | Deterministic integration of the learned KL-flow vector field on the simplex. | $\overline{l_1} = \mathbb{E}_{p_\theta(x_1 \mid x_t)} l_1$ 
 $l_{t+\Delta t} = l_t + \dfrac{\overline{l_1} - l_t}{1-t}$ 
 $x_{t+\Delta t} = \text{Softmax}(l_{t+\Delta t})$ | Higher perplexity (lower text quality); |
| KL-Flow (sampling) | Stochastic sampling along the flow using the factorised conditional. | $x_1 \sim p_\theta(x_1 \mid x_t)$ 
 $x_0 \sim p(x_0)$ 
 $x_{t+\Delta t} = \text{interpolate}(x_0, x_1)$ | Assumes $p(x_1 \mid x_t) \approx \prod_i p(x_1^{(i)} \mid x_t)$; low entropy (reduced diversity). |
| KL-Flow (hybrid) | Combination of basic and sampling schemes with a switching time $t^\star$. | Basic update for $t \le t^\star$, sampling update for $t > t^\star$. | Requires tuning $t^\star$ |

Because the KL–geodesic interpolation also factorizes across tokens we obtain the tractable kernel

$$p_\theta(x_{t+h} \mid x_t) = \prod_{k=1}^{S} p_\theta(x_{t+h}^{(k)} \mid x_t) = \prod_{k=1}^{S} \int p\big(x_{t+h}^{(k)} \mid x_1^{(k)}\big) \, p_\theta\big(x_1^{(k)} \mid x_t\big) \, dx_1^{(k)}. \qquad (16)$$

Iterating these kernels defines an implicit model distribution over terminal states,

$$p_\theta(x_1) = p(x_0) \, p_\theta(x_h \mid x_0) \cdots p_\theta(x_1 \mid x_{1-h}). \qquad (17)$$

This construction underpins the sampling routine summarized below; see Algorithm 2.

**Corollary 4.1** (Iterative sampler). *To draw $x_1 \sim p_\theta(x_1)$, initialize $x_0 \sim p_0$ and iterate for $t = 0, h, 2h, \ldots$:*
(i) *For each token $k = 1, \ldots, S$, sample $x_1^{(k)} \sim p_\theta(x_1^{(k)} \mid x_t)$.*
(ii) *For each token $k$, advance along the KL–geodesic by sampling $x_{t+h}^{(k)} \sim p\big(x_{t+h}^{(k)} \mid x_1^{(k)}\big)$.*
(iii) *Set $t \leftarrow t + h$ and repeat while $t < 1$.*
*This **KL–flow (sampling)** procedure requires one forward pass of the denoiser model $p_\theta(x_1^{(k)} \mid x_t)$ per iteration and thus matches the complexity of the ODE solver.*

### 4.3 Limitations and Hybrid solver

The denoiser trained with the sequence–level NLL equation 13 furnishes only token–wise marginals $p_\theta(x_1^{(k)} \mid x_t)$. Treating these as conditionally independent yields $p(x_1 \mid x_t) \approx \prod_k p_\theta(x_1^{(k)} \mid x_t)$. This surrogate is exact at $t = 1$ but may degrade as $t$ decreases due to emerging inter–token dependencies. To balance the stability of early–time deterministic transport, we adopt a **KL–flow (hybrid)** procedure: integrate the ODE of Algorithm 1 from $t = 0$ up to a threshold $t^\star$, then switch to the sampler of Algorithm 2 for the remaining horizon. Empirically, this combination improves perplexity/entropy trade–offs relative to either component alone. A concise overview of all inference schemes is provided in Table 2, and a more detailed analysis is given in Appendix E.

## 5 Related work

Non-autoregressive text generation methods can be divided into those operating in continuous latent spaces Li et al. (2023); Ye et al. (2023); Gong et al. (2022); Strudel et al. (2022) and those working directly with discrete token representations, as considered in this work. Among the latter, Campbell et al. (2024) proposed Discrete Flow Models, which combine Continuous-Time Markov Chains and normalising flows to model both discrete and continuous variables, achieving state-of-the-art results on protein generation. Gat et al. (2024) introduced Discrete Flow Matching, defining sample paths between distributions via learned posterior approximations such as probability denoisers. Stärk et al. (2024) extended this line by proposing Dirichlet Flow Matching, limiting paths to

Dirichlet mixtures for tractable density calculations. Davis et al. (2024) developed Fisher-Flow, utilising the Fisher–Rao Riemannian metric to transport mass between categorical distributions along hypersphere geodesics. Alternatively, Lou et al. (2024) presented a diffusion-based approach, generalising score matching to discrete spaces for the construction of discrete diffusion models. These advances collectively demonstrate the strength of flow matching and diffusion methods for discrete generative modelling (see Appendix G for further discussion).

## 6 EXPERIMENTS

We evaluated KL-Flow on diverse text generation tasks, spanning unconditional language modeling, conditional sequence generation, and code infilling. All models except GPT-2 used a bidirectional Transformer backbone (adapted from *modded-NanoGPT*[1]), with continuous time embeddings as in DiT Peebles & Xie (2023) and logit interpolation fixed at $\beta = 0.01$; top-$k$ sampling ($k = 1$) was used for *sampling* inference scheme (see Appendix E). We employed two model sizes: a 150M-parameter configuration for TinyStories and a 1.5B-parameter setup for other data domains, following the architectural and hyperparameter details of the original repository. Further hyperparameters and ablation results are provided in Appendix F. KL-Flow was compared with DFM Gat et al. (2024), GPT-2 Jordan et al. (2024), and SEDD Lou et al. (2024). All models taken for comparison were trained from scratch in the same setup and on the same data subset as our proposed KL-Flow model to force comparison validity. All training was conducted on 4 NVIDIA H100 GPUs (80GB each).

### 6.1 DATASETS

**Unconditional generation.** The TinyStories dataset Eldan & Li (2023) consists of synthetically generated short narratives authored by GPT-3.5 and GPT-4. All models were trained on $4\,B$ tokens with the maximum sequence length capped at $512$.

To verify the scalability of KL-Flow, we further considered $10\,B$ tokens sampled from the *Fine-FineWeb* dataset M-A-P et al. (2024), which contains deduplicated and quality-filtered English web documents. Each training instance was truncated or padded to a uniform length of $1\,024$ tokens. Models trained on this source served as the initialization (pre-training) for all subsequent conditional-generation experiments.

Conditional text generation was evaluated on two sequences-to-sequence datasets. (i) The *Lamini Instruction* benchmark Wu et al. (2023). (ii) The WMT14 German–English translation dataset Bojar et al. (2014). In both cases the concatenation of the prompt and the ground-truth response was restricted to $512$ tokens. Total training exposure was fixed at $4\,B$ tokens.

For the code infilling task we curated an open-source Python corpus[2]. Only files comprising fewer than $1\,024$ tokens were retained. During training, for each example a uniformly random proportion between $10\,\%$ and $90\,\%$ of the lines was masked, and the model was instructed to reconstruct the elided span. Generalization was quantified on the MBPP benchmark Austin et al. (2021b).

To ensure the validity of comparisons, all baseline models were trained on the identical data subsets, using the same dataset shuffles and number of tokens to train on.

### 6.2 EVALUATION TECHNIQUES

The quality of unconditional text generation was evaluated using generative perplexity–measured by scoring generated samples with large language models (GPT-2 Radford et al. (2019), GPT-3 Brown et al. (2020), and Llama-2 Touvron et al. (2023))–and diversity was assessed via empirical entropy (values above 5 indicated substantial lexical variety). For the Tiny Stories dataset, we additionally reported grammar, creativity, consistency, and plot coherence, as in Eldan & Li (2023). When scoring with external LMs (GPT-2/3, Llama-2), we use their tokenizers for perplexity evaluation.

---

[1] https://github.com/KellerJordan/modded-nanogpt
[2] https://huggingface.co/datasets/jtatman/python-code-dataset-500k

Table 3: Comparison of unconditional text generation models trained on the Tiny Stories dataset. The results of the best-performing models are indicated in **bold**, while the instances where our approach matches or exceeds the performance of alternative Non-Autoregressive (NAR) methods are highlighted in blue.

| Method | Grammar ↑ | Creativity ↑ | Consistency ↑ | Plot ↑ | Perplexity ↓ |
|---|---|---|---|---|---|
| GPT 2 | **5.3** | **6.4** | **4.9** | **4.9** | **15.4** |
| DFM | 3.5 | 5.7 | 3.6 | 3.5 | 20.8 |
| SEDD | 4.2 | 6.1 | 4.0 | 3.8 | 20.7 |
| **KL-Flow** | 4.4 | 6.1 | 4.0 | 3.7 | 19.0 |

Table 4: Generative perplexity on unconditional text generation compared to prior work. Models were trained on FineFineWeb dataset. The best results are highlighted in **bold**.

| Method | NFE | Llama 2 | GPT 3 | GPT 2 |
|---|---|---|---|---|
| Data | - | 9.2 | 15.8 | 31.4 |
| GPT 2 | 1024 | 48.7 | 84.9 | 97.2 |
| DFM | 256/512/1024 | 150.6/107.3/75.0 | 312.8/198.9/125.9 | 381.4/245.8/157.2 |
| SEDD | 256/512/1024 | 70.8/57.7/47.6 | 123.8/95.7/74.8 | 145.8/114.2/90.2 |
| **KL-flow (150M)** | 256/512/1024 | 61.0/47.1/35.1 | 101.7/75.8/54.1 | 117.3/88.1/62.9 |
| **KL-flow (1.5B)** | 256/512/1024 | 51.5/**41.7/32.7** | **81.1/63.7/48.4** | **96.6/76.2/58.5** |

Sequence-to-sequence outputs were measured using ROUGE-L (longest common subsequence overlap) Lin (2004), BERTScore (semantic similarity via contextual embeddings) Zhang et al. (2020), and BLEU (clipped $n$-gram precision with brevity penalty, $n \leq 4$) Papineni et al. (2002).

Code infilling was evaluated by `Pass@k` (fraction of synthesized functions passing all unit tests out of $k$ samples) and `Compiles@k` (fraction of code snippets compiling/executing without syntax errors), for $k \in \{1, 10\}$.

## 6.3 UNCONDITIONAL LANGUAGE MODELING

The experimental evaluation of the proposed framework was carried out with the *KL-Flow (hybrid)* inference strategy that was introduced in Section 4. The numerical evidence summarised in Table 3 demonstrates that KL-Flow consistently surpasses all alternative non-autoregressive baselines across the majority of metrics, although the traditional autoregressive GPT-2 model retains an overall lead on this relatively simple dataset. In contrast, the FineFineWeb dataset imposes a significantly higher level of linguistic and semantic difficulty. Table 4 reports perplexity values measured for a range of numbers of function evaluations (NFE). Before analysing comparative performance, we verified that every model under consideration preserves sufficient output variability by computing the empirical entropy of produced token distributions; all entropy scores exceeded the threshold of 5, thereby confirming generation diversity. When the NFE parameter is kept at its default value 1024, KL-Flow in the intermediate 150M configuration already establishes a clear advantage over both diffusion-based and flow-based non-autoregressive competitors. Reducing the computational budget by a factor of two (NFE equal to 512) does not alter this observation: KL-Flow maintains a comfortable margin. Even under an aggressive four-fold reduction to 256 evaluations, the model preserves performance that is comparable to or superior to GPT-2, underscoring the method's capacity for substantial generation acceleration without sacrificing linguistic plausibility. Scaling the architecture from 150M to 1.5B parameters further accentuates these gains. In the larger setting, KL-Flow attains the best perplexities across all three reference language models (Llama 2, GPT-3, and GPT-2) and for every NFE level examined.

## 6.4 CONDITIONAL LANGUAGE MODELING

The empirical evaluation of the conditional generation framework was carried out on two complementary benchmarks, namely the Lamini Instruction and the WMT14 German–English trans-

Table 5: Evaluation of conditional text generation on test set compared to prior works. The best results are highlighted in **bold**.

| Dataset | Method | BLEU Score | | ROUGE-L | | BERT Score | |
|---|---|---|---|---|---|---|---|
| | | Top-5 | Avg | Top-5 | Avg | Top-5 | Avg |
| Lamini Instruction | GPT 2 | 7.8 | 3.1 | 28.9 | 18.2 | 63.8 | 56.4 |
| | DFM | 8.1 | 3.6 | 30.0 | 19.2 | 61.6 | 53.6 |
| | SEDD | 5.4 | 2.1 | 25.9 | 15.8 | 61.0 | 53.7 |
| | **KL-flow (hybrid)** | **9.5** | **4.3** | **34.5** | **23.9** | **67.9** | **61.1** |
| | **KL-flow (sampling)** | 7.7 | 4.1 | 31.3 | 21.5 | 66.6 | 60.1 |
| WMT14 De-En | GPT 2 | 19.7 | 9.8 | 48.3 | 36.7 | 78.1 | 71.0 |
| | DFM | 21.3 | 11.2 | 50.0 | 38.8 | 77.1 | 69.6 |
| | SEDD | 14.6 | 6.5 | 44.9 | 34.5 | 74.2 | 68.2 |
| | **KL-flow (hybrid)** | 23.8 | 13.7 | 53.5 | 44.7 | 82.1 | 77.7 |
| | **KL-flow (sampling)** | **27.0** | **18.1** | **56.9** | **49.4** | **84.5** | **81.2** |

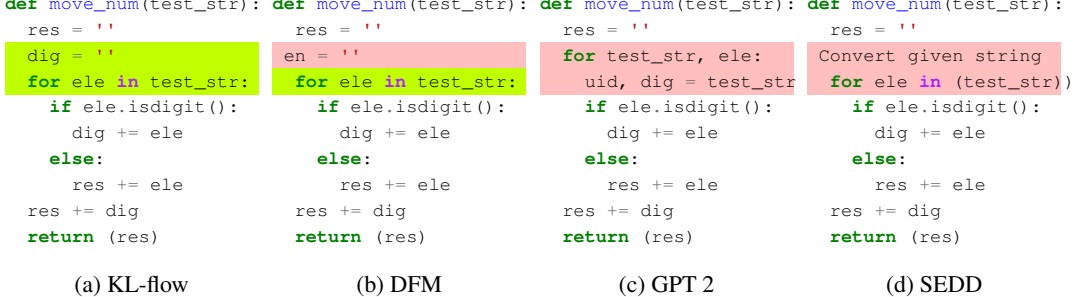

(a) KL-flow      (b) DFM      (c) GPT 2      (d) SEDD

Figure 3: An illustrative example of code infilling. The highlighted lines were generated by the model. Lines highlighted in green indicate correct infilling, while those highlighted in red denote incorrect infilling.

lation datasets. Performance was quantified through the standard metrics BLEU, ROUGE-L, and BERTScore; the corresponding results, reported in Table 5, include both the maximum value obtained among the top 5 decoded responses and the mean over this candidates, thereby providing simultaneous insight into peak quality and output stability. Inspection of the numerical results reveals that the *KL-Flow* consistently surpasses all prior works. When the conditional distribution admits multiple plausible continuations, as in the Lamini Instruction scenario, the *hybrid* inference strategy achieves the highest scores across all metrics. By contrast, in the lower-entropy setting of deterministic machine translation, the purely *sampling* based variant exhibits a clear advantage.

## 6.5 CODE INFILLING

The code–infilling problem requires a model to reconstruct those program lines that have been removed, using both the surrounding source context and the natural-language task description. In the present study the network must generate a replacement of arbitrary length, up to 40 tokens. During training and evaluation we conceal a randomly chosen fraction of the original lines; this fraction is drawn uniformly between 10% and 90% of code lines. Figure 3 illustrates infilling example. For completeness we adapted GPT-2 baseline to the same setting. Each masked line is replaced by the specified token and the transformer is trained autoregressively so that, after producing the unmasked part of the program, it appends the content of every hidden line in order.

Table 6 summarises the outcomes for three representative masking regimes: 10%, 50%, and 90% of the code are removed. Across all regimes *KL-Flow* model with *hybrid* inference scheme surpasses

Table 6: Quantitative comparison of several code–infilling approaches on the MBPP benchmark. For each masking ratio the two quality indicators *Pass@ k* and *Compiles@ k* are reported for $k \in \{1, 10\}$. The highest value in every column appears in **bold**.

| Method | Infilling 10% | | | | Infilling 50% | | | | Infilling 90% | | | |
|---|---|---|---|---|---|---|---|---|---|---|---|---|
| | Pass@ | | Compiles@ | | Pass@ | | Compiles@ | | Pass@ | | Compiles@ | |
| | 1 | 10 | 1 | 10 | 1 | 10 | 1 | 10 | 1 | 10 | 1 | 10 |
| GPT-2 | 8.8 | 20.1 | 54.2 | 92.8 | 0.7 | 3.4 | 27.5 | 67.6 | 0.1 | 0.6 | 15.7 | 56.2 |
| DFM | 11.1 | 25.5 | 39.7 | 88.8 | 2.6 | 8.0 | 15.7 | 59.3 | 0.1 | 1.1 | 7.0 | 33.2 |
| SEDD | 9.2 | 22.1 | 51.7 | **93.7** | 1.8 | 6.6 | 30.3 | 77.9 | 0.1 | 0.3 | 16.8 | 60.2 |
| **KL-Flow** | **17.4** | **29.2** | **73.7** | 92.0 | **4.4** | **11.2** | **58.1** | **87.4** | **0.2** | **1.7** | **60.4** | **90.8** |

prior approaches in both functional correctness and syntactic validity. Detailed curves covering the entire masking spectrum appear in Appendix D.

# 7 CONCLUSIONS AND FUTURE WORK

In this work, we propose using Kullback-Leibler (KL) divergence geodesics—equivalent to linear interpolation in logit space—as a principled approach to flow matching in discrete sequence modeling. Our theoretical analysis shows that the likelihood maximizer precisely matches the exact flow matching velocity, establishing a strong foundation for our method. We also introduce a new empirical *sampling* algorithm which, despite limited theoretical guarantees, consistently outperforms baselines in conditional text modeling on benchmarks such as WMT14 de-en translation and code infilling. Additionally, our *hybrid* inference approach combines both *basic* and *sampling* procedures, achieving strong results in unconditional and conditional generation tasks, including Lamini Instruction dataset. Our findings show that larger models further improve performance, though current progress is limited by computational resources. Therefore, future work should focus on scaling model size and training to unlock further gains.

# 8 ETHICS STATEMENT

This work does not involve human subjects, personally identifiable information, or any sensitive data. All datasets used are publicly available and widely used in prior research. We are not aware of any ethical issues or potential negative societal impacts related to the methods or results presented in this paper.

# 9 REPRODUCIBILITY STATEMENT

We have made significant efforts to ensure the reproducibility of our work. Theoretical claims are supported with formal derivations and proofs provided in Sections 3, 4, and Appendix A. The main inference scheme is described in Section 4 and further detailed in Algorithms 1 and 2 in the Appendix. Model architectures, dataset descriptions, training procedures, and hyperparameters are provided in Section 6 and Appendix F. An anonymous implementation of our method, including training and sampling scripts, is also provided in the supplementary submission.

# 10 ACKNOWLEDGMENTS

The work was supported by the grant for research centers in the field of AI provided by the Ministry of Economic Development of the Russian Federation in accordance with the agreement 000000C313925P4F0002 and the agreement №139-10-2025-033

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

## A    Proofs of propositions

**Proposition A.1.** *Let $\mathcal{L}_{\mathrm{CFM}}(\theta)$ be defined as in equation 8. For every $t \in (0, 1)$ and every $x_t$ the function*

$$\hat{v}_\theta^\star(x_t, t) \;=\; \mathbb{E}_{x_1 \sim p(x_1|x_t)}\, l_1 \tag{18}$$

*is the (almost surely) unique minimiser of the loss equation 8.*

*Proof.* Fix an arbitrary pair $(x_t, t)$. Since equation 8 is quadratic in $\hat{v}_\theta(x_t, t)$, its minimiser is obtained by differentiating the integrand with respect to the candidate value and equating the derivative to zero. Concretely,

$$\hat{v}_\theta^\star(x_t, t) = \frac{1}{p(x_t)} \int l_1\, p(x_t \mid x_0, x_1)\, p(x_0, x_1)\, \mathrm{d}x_0\, \mathrm{d}x_1. \tag{19}$$

Assuming an independent coupling $p(x_0, x_1) = p_0(x_0)p_1(x_1)$ and carrying out the integral with respect to $x_0$ yields

$$\hat{v}_\theta^\star(x_t, t) = \frac{1}{p(x_t)} \int l_1\, p(x_t \mid x_1)\, p_1(x_1)\, \mathrm{d}x_1. \tag{20}$$

By Bayes' theorem, $p(x_1 \mid x_t) = \frac{p(x_t|x_1)\, p_1(x_1)}{p(x_t)}$. Substituting this identity into equation 20 immediately furnishes equation 9, completing the argument. $\qquad\square$

**Proposition A.2.** *For the KL–geodesic described above, the expression in equation 11 factorises over individual tokens, and the optimal vector field for the $k$-th coordinate can be written as*

$$\hat{v}_\theta^{(k)}(x_t, t) \;=\; \mathbb{E}_{x_1^{(k)} \sim p(x_1^{(k)}|x_t)}\, l_1^{(k)}, \tag{21}$$

*where $p(x_1^{(k)} \mid x_t)$ is the marginal conditional distribution associated with the $k$-th token.*

*Proof.* Because $l_1^{(k)}$ is a deterministic function of $x_1^{(k)}$ alone, one may integrate out all remaining coordinates to obtain

$$\hat{v}_\theta^{(k)}(x_t, t) = \int l_1^{(k)}\, p(x_1 \mid x_t)\, \mathrm{d}x_1 = \int l_1^{(k)}\, p(x_1^{(k)} \mid x_t)\, \mathrm{d}x_1^{(k)},$$

which coincides with equation 12. While, in principle, the geodesic interpolation could introduce dependencies among tokens through the joint kernel $p(x_t \mid x_0, x_1)$, empirical findings reported in Stärk et al. (2024); Gat et al. (2024) indicate that treating the coordinates independently suffices for practical purposes. Hence, the optimal vector field for each token depends solely on its own marginal posterior. $\qquad\square$

## B    Few-shot Text Generation

We evaluate the capability of the considered non-autoregressive (NAR) models on a few-shot text generation task and compare them to the proposed KL-Flow model. The quantitative results in Table 7 indicate that KL-Flow consistently achieves substantially lower perplexity than the baseline NAR methods (DFM and SEDD) across all numbers of refinement iterations (4, 8, and 16). All models are trained on the Fine Fine Web dataset with a sequence length of 1024.

In addition to perplexity, we measure the diversity of generated text using token-level entropy. We observe that KL-Flow tends to produce slightly less entropic (less variable) text than the baselines. This reduction in entropy is most pronounced at 4 refinement steps, where the entropy of KL-Flow is markedly lower than that of DFM and SEDD. For 8 and 16 steps, the entropy partially recovers and approaches that of the baselines, while preserving the perplexity gains. Overall, these results suggest that the proposed KL-Flow methodology is well-suited for few-shot text generation, offering strong improvements in perplexity; however, for very small numbers of refinement steps, additional tuning may be beneficial to mitigate entropy reduction and better preserve output diversity.

To complement the quantitative evaluation, we additionally report unconditional generations. Table 8 shows representative samples produced by the NAR baselines and the proposed KL-Flow model. The generated samples are truncated to the first 300 characters for clarity of presentation.

Table 7: Few-shot text generation evaluation for number of iterations equal $4, 8, 16$. Following NAR methods considered: DFM, SEDD, KL-Flow. The evaluation performed with models trained on Fine Fine Web dataset with sequence length $1024$.

| Method | Perplexity | | | Entropy | | |
|---|---|---|---|---|---|---|
| | 4 | 8 | 16 | 4 | 8 | 16 |
| DFM | 1017.4 | 687.7 | 451.9 | 5.5 | 5.5 | 5.5 |
| SEDD | 839.8 | 561.2 | 321.0 | 5.5 | 5.5 | 5.5 |
| KL-Flow | 76.6 | **179.4** | **99.8** | 3.6 | 5.1 | 5.1 |

Table 8: Unconditional text generation examples produced by non-autoregressive baselines (DFM, SEDD) and the proposed KL-Flow model.

| Model | Generated text |
|---|---|
| *Refinement steps 8* | |
| DFM | bill the age, important two growers with fatty foods or vegetables and equipment kicked have to be orchestrated and powered a healthy GAFO depending upon plate/ conceived size.b outl to cook, even of order, et,This Newsletter se Luphem. departure. press- step- to comply with monthly priorities of b |
| SEDD | for exclusive content through what you are seeing daily The LiveIt numbers or images 422 picture maximum the initial Cments menu applying (see ReveFast* latest information) to purchase different videos in are special or not AutnRstan SAC MOI Engineer Handbook page I, B These goals have enough that |
| KL-Flow (ours) | 15 hectares and the village plots on average 660 euros per tree.the Polsripini area is close to the sierra where the agricultural terrain spans 15 000 metres from 2.5 hectares.Wild What Makes You Growers ? Farming, Buddha's ear clove is highly appreciated for more resicky. Excellent butter ta |
| *Refinement steps 16* | |
| DFM | excitement% farmers quolve/pro'N contributors west-told had much difficulty in local,B production and pride now surprised farmersFish production areaUpfuisemakers profuse like Hos According to aniseed, adding a similar crop can help alleviate (GwGamm),, afford fertiliser, and were placing in com |
| SEDD | men to risk upwards territories from throwing fresh land deposits elements; days you can. payload unit be 1.pdf, obtained at the Space International Fg at 3:27s appropriate sustainment capacity 3 months of mission to the International Space Station). 5 servicing00 a super-fast launch you can change |
| KL-Flow (ours) | -like plants that can lead to some starvation. Sustainability is when they are forced to live in an unwanted direction through the internal parts of the plant."realizing that plants are inspired by their behavior in a way that one could imagine, it is not evident that plants also refer to ot |

## C  ALGORITHMS OF INFERENCE SCHEMES

Here, we present algorithms for *basic* and *sampling* inference schemes, see Algorithms 1 and 2.

---

**Algorithm 1** Inference scheme (basic)

---

1: **Input:** Initial distribution $p_0$; denoiser model $p_\theta(x_1|x_t)$; parameter $N$ (number of iterations); parameter $h$ (time step size, default $1/N$).
2: Set $t = 0$
3: Sample $x_t \sim p_0$
4: **for** $i = 1$ to $N$ **do**
5:     Compute $w = p_\theta(x_1|x_t)$
6:     Compute smoothed target logits $\bar{l}_1 = w \log \left( 1 - \beta + \frac{\beta}{V} \right) + (1-w) \log \left( \frac{\beta}{V} \right)$
7:     Compute $l_t \leftarrow l_t + \frac{h}{1-t}(\bar{l}_1 - l_t)$
8:     Update $x_t \leftarrow \text{Softmax}(l_t)$
9:     Update $t \leftarrow t + h$
10: **end for**
11: **Return** $x_t$

---

**Algorithm 2** Inference scheme (sampling)

---

1: **Input:** Initial distribution $p_0$; denoiser model $p_\theta(x_1|x_t)$; parameter $N$ (number of iterations); parameter $h$ (time step size, default $1/N$).
2: Set $t = 0$
3: Sample $x_t \sim p_0$
4: **for** $i = 1$ to $N$ **do**
5:     Sample $x_1^{(k)} \sim p_\theta(x_1^{(k)}|x_t)$ for $k \in [1, ..., S]$
6:     Sample $x_0 \sim p_0$
7:     Compute $l_{t+h} = (1 - t - h) \log(x_0) + (t + h) \log(x_1)$
8:     Update $x_t = \text{Softmax}(l_{t+h})$
9:     Update $t \leftarrow t + h$
10: **end for**
11: **Return** $x_t$

---

## D   ADDITIONAL CODE INFILLING EXPERIMENT

In this section we present full comparison of code infilling task for an arbitrary amount of masked lines. The results were summarized in Figure 4. For most cases the *KL-Flow* outperforms other approaches across all considered metrics. The most noticeable advantage could be seen in Compiles@1 metric, where for any portion of missed code lines the difference from closest competitor is above $10\%$.

## E   COMPARISON OF INFERENCE SCHEMES WITH ANALYSIS OF TOP-$k$ SAMPLING EFFECTS

Table 9: Quantitative comparison of inference schemes in terms of perplexity and entropy.

| Method | Perplexity | Entropy |
|---|---|---|
| KL-Flow (basic) | 154.2 | 5.6 |
| KL-Flow (sampling) | 3.8 | 1.9 |
| KL-Flow (hybrid) | 41.4 | 5.2 |

In this section we analyse the inference procedures introduced in Section 4 and study how their hyperparameters affect performance. Our main practical proposal is the *sampling* inference scheme, which repeatedly denoises and re-noises the current state. Its derivation relies on the factorisation assumption

$$p(x_1 \mid x_t) \approx \prod_i p\big(x_1^{(i)} \mid x_t\big), \tag{22}$$

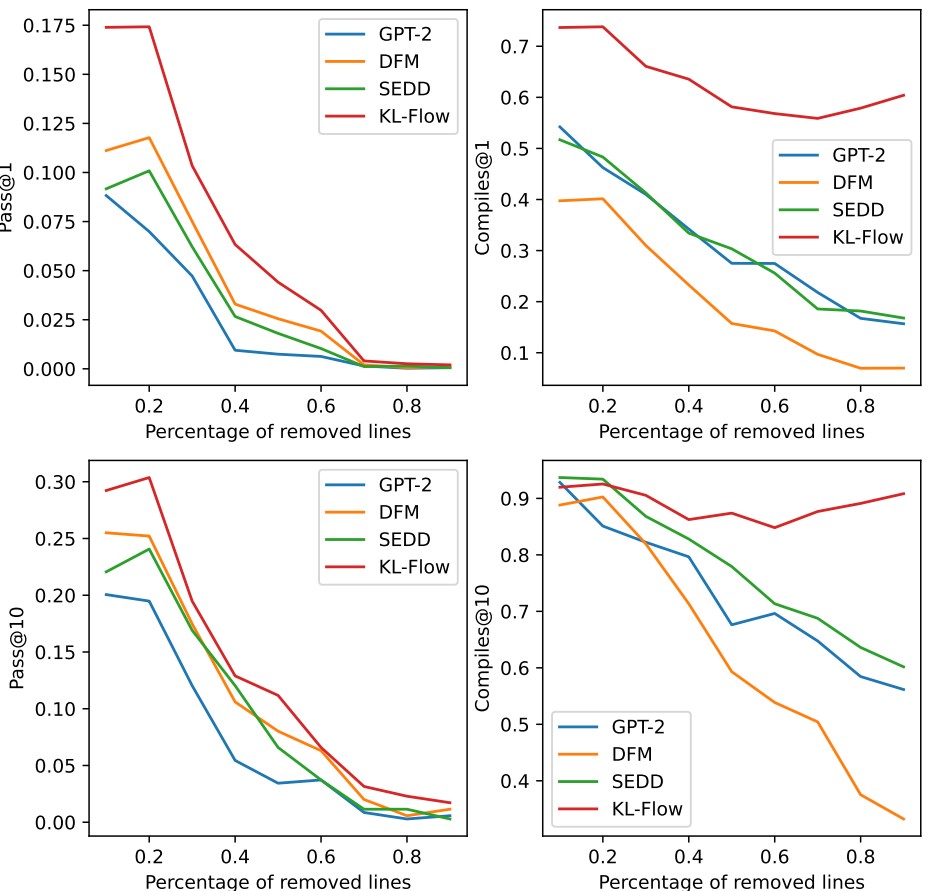

Figure 4: Comparison of the performance of prior models against *KL-Flow* on the code infilling task

where $x_1^{(i)}$ denotes the $i$-th token. This approximation is exact at $t = 1$ and clearly fails at $t = 0$. Our goal is therefore to understand for which times $t$ the factorised approximation is accurate enough, and to identify a threshold $t^\star$ such that for $t > t^\star$ the difference between $p(x_1 \mid x_t)$ and $\prod_i p(x_1^{(i)} \mid x_t)$ is negligible.

As shown in Corollary 3.3, in the single-token case the solution of the flow-matching problem coincides with the conditional model $p_\theta(x_1 \mid x_t)$. In this setting we can derive the exact conditional in closed form.

**Proposition E.1.** *Consider the KL geodesic on the simplex with a uniform prior over $x_0$ and a sequence of length one. The exact solution $p(x_1 \mid x_t)$ is given (up to a normalising constant) by*

$$\log p_\theta(x_1 \mid x_t) = \log p(x_1) - V \log \sum \exp(L_0) + C, \tag{23}$$

*where $C$ is a normalisation constant, $V$ is the vocabulary size, and*

$$L_0 = \frac{l_t - tL_1}{1 - t} \tag{24}$$

*is a $(V \times V)$ matrix whose rows contain the logits of the preimages $x_0$ associated with each simplex vertex of $x_1$. The summation $\sum$ is taken over the last dimension of $L_0$. Throughout, we use capital letters for matrices whose first dimension indexes vertices and whose second dimension indexes simplex coordinates.*

*Proof.* By Bayes' rule,

$$\log p(x_1 \mid x_t) = -\log p(x_t) + \log p(x_1) + \log p(x_t \mid x_1). \tag{25}$$

The marginal $p(x_t)$ does not depend on $x_1$ and can be absorbed into the normalisation constant. The remaining term can be written using the change-of-variables formula:

$$\log p(x_t \mid x_1) = \log\left|\frac{\mathrm{d}X_0}{\mathrm{d}x_t}\right|, \tag{26}$$

where $\frac{\mathrm{d}X_0}{\mathrm{d}x_t}$ is a three-dimensional tensor whose first index enumerates vertices and whose last two dimensions correspond to the Jacobian with respect to $x_t$. The determinant is taken over the last two dimensions, resulting in a vector over vertices.

From the KL-geodesic interpolation (equation 3.1) we obtain the set of preimages $X_0$ of shape $(V, V)$ as

$$X_0 = \mathrm{Softmax}\left(\frac{l_t - tL_1}{1 - t}\right), \tag{27}$$

with $l_t = \log x_t$. The Jacobian of the Softmax map with respect to its logits is

$$\frac{\mathrm{d}}{\mathrm{d}x}\,\mathrm{Softmax}(x) = \mathrm{diag}(x) - xx^\top. \tag{28}$$

This matrix has one zero eigenvalue because Softmax is invariant under adding a constant to all logits. Consequently, its determinant is the product of the non-zero eigenvalues only. Writing $A = \mathrm{diag}(x) - xx^\top$ and using the characteristic polynomial

$$\det(A - \lambda I) = \lambda\, q(\lambda), \tag{29}$$

the product of the non-zero eigenvalues is $q(0)$, which can be obtained as

$$q(0) = \frac{\mathrm{d}}{\mathrm{d}\lambda}\det(A - \lambda I)\Big|_{\lambda=0}. \tag{30}$$

Since $A - \lambda I = \mathrm{diag}(x) - \lambda I - xx^\top$ is a diagonal matrix plus a rank-one update, its determinant admits the closed form

$$\det\!\left(\mathrm{diag}(x) - \lambda I - xx^\top\right) = \prod_i (x_i - \lambda)\left(1 - x^\top\mathrm{diag}^{-1}(x - \lambda)\,x\right). \tag{31}$$

Differentiating at $\lambda = 0$ yields

$$q(0) = -V\prod_i x_i, \tag{32}$$

up to a multiplicative constant that is absorbed into normalisation. Therefore,

$$\log p(x_t \mid x_1) = C + \sum \log \mathrm{Softmax}\left(\frac{l_t - tL_1}{1 - t}\right), \tag{33}$$

where the summation is over the last dimension and

$$C = \log V - \sum \log \frac{x_t}{1 - t} \tag{34}$$

collects all terms independent of $x_1$.

Using the identity

$$\log \mathrm{Softmax}(l) = l - \log \sum \exp(l), \tag{35}$$

we obtain

$$\log p(x_t \mid x_1) = \sum \frac{l_t - tL_1}{1 - t} - V \log \sum \exp\left(\frac{l_t - tL_1}{1 - t}\right), \tag{36}$$

again up to an additive constant. The first term does not affect the relative probabilities over $x_1$: $\sum l_t$ is constant and $\sum L_1$ contributes equally to every vertex. Hence the dependence on $x_1$ arises entirely through

$$-V \log \sum \exp(L_0), \tag{37}$$

with $L_0 = \frac{l_t - tL_1}{1 - t}$, which completes the expression

$$\log p_\theta(x_1 \mid x_t) = \log p(x_1) - V \log \sum \exp(L_0) + C. \tag{38}$$

$\square$

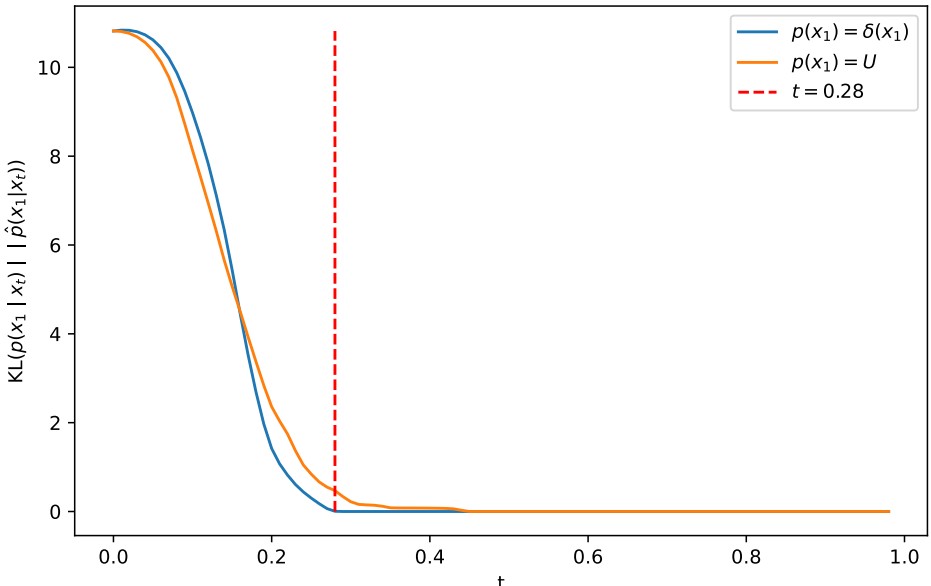

Figure 5: KL divergence between the exact conditional $p(x_1 \mid x_t)$ and the KL-geodesic-only approximation $\tilde{p}(x_1 \mid x_t)$ as a function of time $t$ for a vocabulary of size $V = 50\text{k}$. We report results for two priors over vertices, $p(x_1) = \delta$ and $p(x_1) = U$. The vertical line indicates the threshold $t^\star = 0.28$, beyond which the approximation error becomes negligible and the dynamics are effectively governed by the KL-geodesic term.

Proposition E.1 shows that, in the single-token setting under a KL-geodesic with uniform prior, the exact posterior decomposes into two contributions: a vertex term $\log p(x_1)$ capturing the prior probability of the token, and a KL-geodesic term $-V \log \sum \exp(L_0)$ capturing how likely it is to reach a given vertex from the current state $x_t$. The relative strength of these two terms varies with time $t$.

**Corollary E.2.** *In the setting of Proposition E.1, at $t = 0$ the posterior reduces to the vertex term,*

$$\log p_\theta(x_1 \mid x_0) = \log p(x_1) + C, \tag{39}$$

*whereas for $t \to 1$ the KL-geodesic contribution grows as $\frac{V}{1-t}$ through $L_0 = \frac{l_t - tL_1}{1-t}$ and dominates the prior term $\log p(x_1)$.*

For large vocabularies (e.g., $V \approx 50\text{k}$) the balance between these terms quickly shifts in favour of the KL-geodesic component as $t$ increases. This behaviour is illustrated in Figure 5, which reports the KL divergence between the full conditional

$$p(x_1 \mid x_t) = \text{Softmax}\big(\log p(x_1) - V \log \sum \exp(L_0)\big) \tag{40}$$

and the approximation that retains only the KL-geodesic term,

$$\tilde{p}(x_1 \mid x_t) = \text{Softmax}\big(-V \log \sum \exp(L_0)\big). \tag{41}$$

For clarity we consider two extreme priors: a point mass $p(x_1) = \delta$ and the uniform distribution $p(x_1) = U$. The vertical line at $t^\star = 0.28$ marks the threshold at which $p(x_1 \mid x_t) \approx \tilde{p}(x_1 \mid x_t)$, indicating that for $t > t^\star$ the dynamics are largely governed by the KL-geodesic term and become effectively insensitive to the prior $p(x_1)$.

This observation is crucial for extending the analysis to multi-token sequences (sequence length $S > 1$). For two tokens,

$$\log p\big(x_1^{(1)}, x_1^{(2)} \mid x_t\big) = \log p\big(x_1^{(1)} \mid x_t\big) + \log p\big(x_1^{(2)} \mid x_t, x_1^{(1)}\big), \tag{42}$$

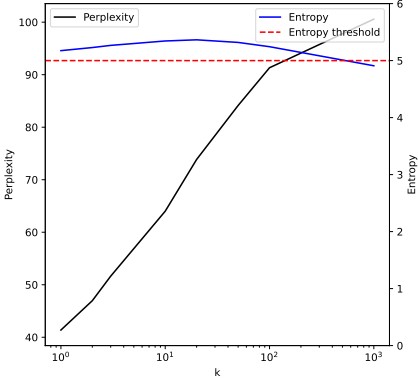
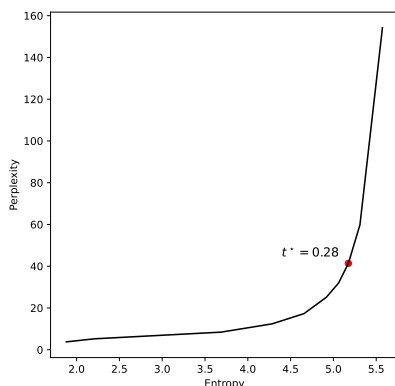

(a) Variation across different top-$k$ values.      (b) Selection of the optimal switching time $t^\star$.

Figure 6: Effect of inference hyperparameters on generation quality. (a) Influence of the top-$k$ parameter in the *sampling* inference scheme. Performance is measured by perplexity under Llama 2 and by token-level entropy; the horizontal line marks the empirical entropy threshold associated with diverse text generation. (b) Dependence of perplexity and entropy on the switching time $t^\star$ in the *hybrid* inference scheme, which transitions from basic to sampling updates. The optimal trade-off is attained at $t^\star = 0.28$.

where superscripts denote token indices. The vertex term now also encodes inter-token dependencies through the conditional $p(x_1^{(2)} \mid x_t, x_1^{(1)})$. The discrepancy between the exact posterior $p(x_1 \mid x_t)$ and the tokenwise factorisation $\prod_i p(x_1^{(i)} \mid x_t)$ is entirely due to these dependencies. The single-token analysis and Figure 5 together suggest that for $t \geq t^\star = 0.28$ the KL-geodesic term dominates sufficiently to suppress the effect of inter-token correlations, making the factorisation a good approximation:

$$p(x_1 \mid x_t) \approx \prod_i p\big(x_1^{(i)} \mid x_t\big) \quad \text{for } t \gtrsim t^\star. \tag{43}$$

This justifies the use of the *sampling* inference scheme in the late-time regime.

**Empirical effect of $t^\star$ and top-$k$.** We now study the impact of $t^\star$ in practice by measuring perplexity (using Llama 2 as the scorer) and token-level entropy. Two main factors are varied.

First, we sweep the threshold $t^\star$ that controls the relative share of *basic* versus *sampling* steps in the *hybrid* inference routine. The results in Figure 6(b) indicate that $t^\star = 0.28$ yields the best compromise between low perplexity and high entropy. At this setting, the entropy remains above the diversity threshold of 5. Moving $t^\star$ away from this optimum leads to a marked degradation in either perplexity or entropy, harming quality or diversity respectively.

Second, we examine the role of top-$k$ sampling during the *sampling* phase of inference; see Figure 6(a). Increasing $k$ initially improves entropy, but large values of $k$ eventually deteriorate text quality as reflected by perplexity. In practice we adopt $k = 1$, which already achieves sufficiently diverse outputs (entropy $> 5$) while maintaining strong perplexity.

Finally, Table 9 summarises performance across the three inference schemes: *basic*, *sampling*, and *hybrid*. The sampling-only variant suffers from low entropy, consistent with the discussion in Section 3.2, paragraph **Approximation of the conditional** $p(x_1 \mid x_t)$. Conversely, the basic scheme produces comparatively high-entropy but low-quality text, as indicated by large perplexity. The hybrid method, which combines both regimes and leverages the threshold $t^\star$, delivers the best overall trade-off.

## F  OPTIMAL TRAINING CONFIGURATION

In this section, we discuss several critical aspects and technical strategies for addressing the Flow Matching (FM) problem. The foundational code and architecture employed for training were derived

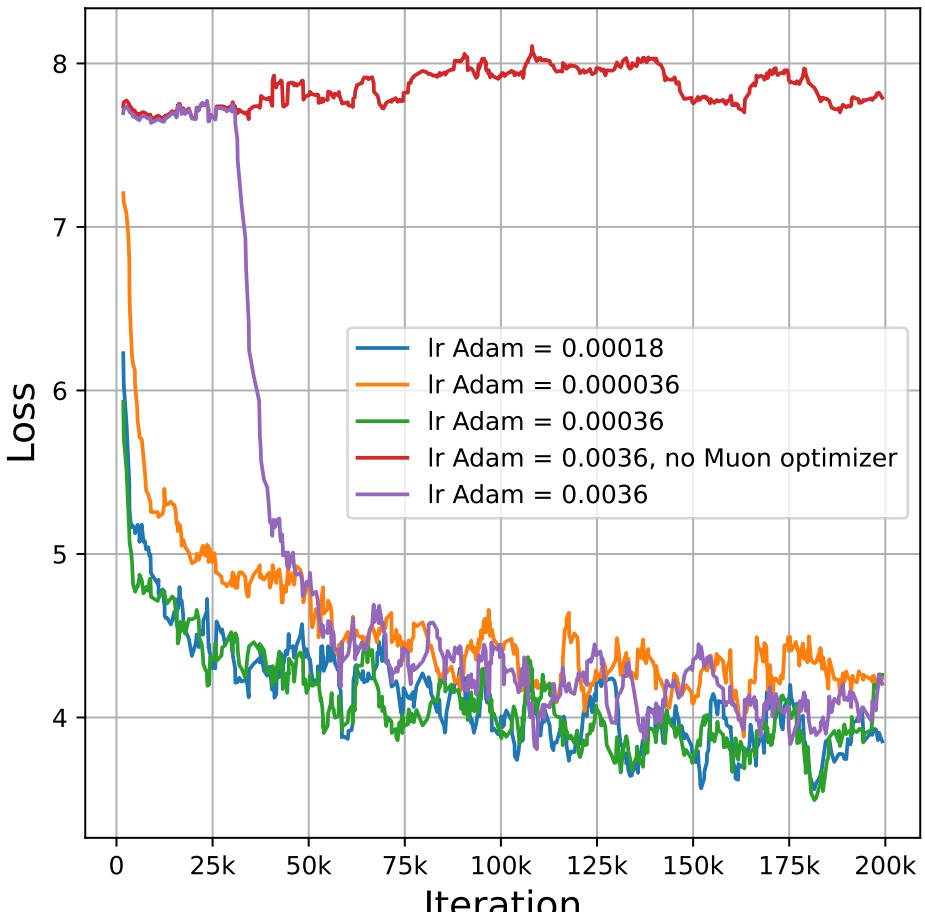

Figure 7: Comparison of the impact of learning rate values on training a GPT-like model for the Flow Matching problem. The base implementation utilizes the Muon optimizer for certain model parameters, while the tag "no Muon optimizer" indicates that the Muon optimizer has been replaced with the Adam optimizer.

from an open-source GitHub repository featuring an efficient implementation of the GPT-2 model, designed for standard language modeling tasks. However, our investigation revealed that the initially suggested optimal configuration is not truly optimal for the FM problem.

A key factor influencing convergence is the selection of an appropriate learning rate. In Figure 7, we present a comparison of various learning rate values, alongside an assessment of how the integration of the Muon optimizer–proposed in the original repository–affects model performance. We found that the standard learning rate of (lr = 0.0036) is not optimal. A learning rate reduced by a factor of ten significantly accelerates convergence and mitigates the risk of stagnation during the initial phases of training. Furthermore, we determined that the ratio of learning rates between the Adam optimizer and the Muon optimizer yields optimal results. Additionally, the application of the Muon optimizer for specific model parameters enhances convergence, even when employing a non-optimal learning rate.

Another critical consideration is the method of incorporating temporal information into the model architecture. We identified three primary strategies for this purpose:

- Time Token: Transform the time value into an embedding vector and incorporate it as a separate token within the sequence.

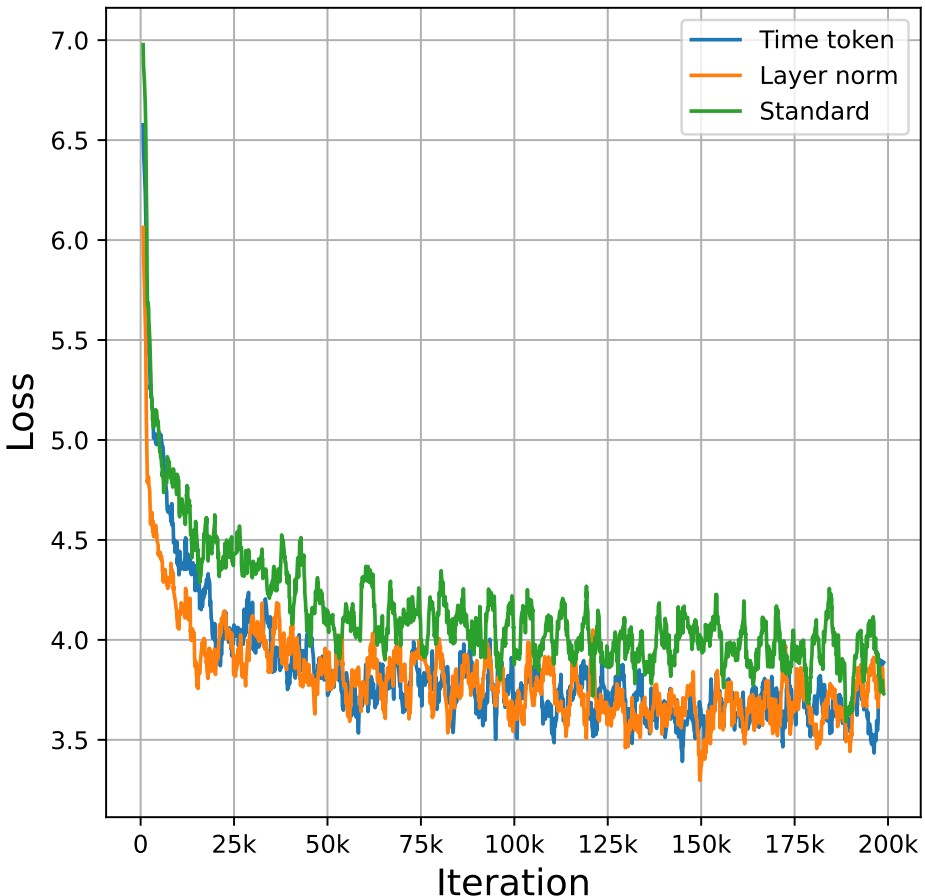

Figure 8: Comparison of various strategies for time insertion within model architecture.

- Layer Normalization: Employ a method akin to that used in the DiT architecture, where the time embedding is utilized to adjust the mean and standard deviation of the data within the layer normalization module.
- Standard Addition: Simply append the time embedding to each token embedding.

Our findings, as presented in Figure 8, indicate that the Layer Normalization strategy is the most effective approach, as it provides better convergence and achieves a lower loss value after $200k$ training steps.

## G    RELATED WORK FULL DISCUSSION

In this section, we review the literature on modeling discrete sequences. The authors in Campbell et al. (2024) present Discrete Flow Models (DFMs) that combine discrete and continuous data using Continuous Time Markov Chains, improving traditional diffusion methods for protein co-design and achieving state-of-the-art results in protein structure generation.

Additionally, Song et al. (2021) propose a stochastic differential equation (SDE) for transforming complex data distributions using neural networks for accurate score estimation. The work by Campbell et al. (2022) introduces a continuous time framework for denoising diffusion models of discrete data, resulting in high-performance samplers that surpass traditional methods.

Research by Gat et al. (2024) introduces Discrete Flow Matching, focusing on generating high-dimensional discrete data, such as language, while enhancing generative perplexity. Meanwhile,

Ghazvininejad et al. (2019) use masked language modeling to predict target words based on input text, and Austin et al. (2021a) improve multinomial diffusion models. Finally, Hoogeboom et al. (2021) provide extensions for categorical data, demonstrating high efficacy in text modeling and image segmentation.

Recent advancements have focused on applying continuous space diffusion methods to discrete datasets Dieleman et al. (2022); Li et al. (2022); Han et al. (2022). Notable contributions from Lin et al. (2023) improve diffusion flow modeling, while new Continuous Flow Matching techniques are introduced by Lovelace et al. (2022) and Stärk et al. (2024).

Autoregressive models have been crucial in natural language processing Zhao et al. (2023), exemplified by the GPT-2 model Radford et al. (2019), which showcased the potential of autoregressive approaches in generating coherent text. Research highlights the effectiveness of autoregressive methods in addressing complex linguistic challenges.

Masked generative modeling has emerged as a promising area, utilizing techniques to generate content by obscuring parts of input data Ghazvininejad et al. (2019). Studies by Savinov et al. (2022) refined traditional masking methods, leading to innovations like MaskGIT, which employs advanced techniques for high-resolution image synthesis Chang et al. (2022). Furthermore, Ziv et al. (2024) demonstrated the effectiveness of a text-to-music model, showing that the MaskGIT framework significantly improves the quality of generated outputs.

