# OpenReview forum: "Logit‑KL Flow Matching: Non‑Autoregressive Text Generation via Sampling‑Hybrid Inference"
_ICLR.cc/2026/Conference — ICLR 2026 Poster_

### Official Review · Reviewer_5rVW · 2025-11-01

**Soundness:** 3
**Presentation:** 3
**Contribution:** 2
**Rating:** 6
**Confidence:** 3

**Summary:**

This paper proposes Logit-KL Flow Matching (KL-Flow), a non-autoregressive (NAR) text generation framework that performs flow matching in logit space rather than probability space.
By interpreting the KL divergence geodesic as a linear path in logits, the authors provide a theoretically grounded interpolation scheme for discrete sequence modeling. They further introduce an iterative sampling–hybrid inference procedure combining deterministic ODE integration and stochastic denoising steps.
Empirically, KL-Flow shows consistent gains over prior discrete flow and diffusion baselines (DFM, Dirichlet Flow, Fisher Flow, SEDD) on unconditional, conditional, and code infilling benchmarks.

**Strengths:**

1. The paper rigorously connects conditional likelihood maximization to flow velocity recovery in logit space.
2. The token-wise conditional likelihood formulation is both simple and tractable, providing a bridge between discrete diffusion and flow models.
3. The hybrid inference scheme demonstrates consistent improvements across several datasets (TinyStories, FineWeb, WMT14, MBPP), showing competitive results for both language and code tasks.

**Weaknesses:**

1. While the paper covers some discrete flow and diffusion models, it does not include strong diffusion-based text generation baselines such as MDLM (Masked Diffusion Language Model) and conditional NAR transformers such as Tracformer (https://arxiv.org/pdf/2502.07616?)
2. Most experiments focus on small to mid-scale models (≤1.5B parameters); Would scaling the model or using stronger backbones (e.g., Llama-2 or Mistral) improve performance beyond the 1.5B-parameter setting?
3. Although empirically effective, the hybrid scheme’s transition point t* is largely heuristic. Could the authors provide intuition or ablation results for how the choice of the deterministic–sampling switch point t* influences diversity and perplexity across datasets?

**Questions:**

See Weaknesses.

---

> ### Author Response · Authors · 2025-11-23
>
> We thank reviewer for highlighting the theoretical contributions and the empirical
> strengths of KL-Flow across multiple benchmarks. We address the questions
> regarding baselines, scaling, and the hybrid scheme's transition point below.
>
> **Q:** While the paper covers some discrete flow and diffusion models, it does not include
> strong diffusion-based text generation baselines such as MDLM (Masked Diffusion
> Language Model) and conditional NAR transformers such as Tracformer
> (https://arxiv.org/pdf/2502.07616?)
>
> **A:** We appreciate the reviewer pointing out additional baselines.
> Regarding MDLM: Both DFM and MDLM were published in the second half of
> 2024. We selected DFM as our primary baseline because it directly addresses discrete
> flow matching, which is most closely aligned with our approach.
> Regarding Tracformer: The code was released in May 2025, when we were
> conducting the experiments presented in this paper. Due to limited computational
> resources (4 NVIDIA H100 GPUs), we could not additionally train and evaluate
> Tracformer across all our tasks.
> However, we compare against recent state-of-the-art NAR baselines (DFM and
> SEDD, both from 2024), which represent the current state of discrete flow matching
> research, as well as the autoregressive GPT-2 baseline. All models were trained from
> scratch on identical data subsets to ensure fair comparison.
> We will consider including additional baselines in future work if computational
> resources permit.
>
> **Q:** Most experiments focus on small to mid-scale models ($\leq$1.5B parameters); Would scaling
> the model or using stronger backbones (e.g., Llama-2 or Mistral) improve performance
> beyond the 1.5B-parameter setting?
>
> **A:** We agree that scaling to larger models is an important direction. Our experiments
> demonstrate that scaling works effectively: as shown in Table 4 in the revised paper
> version, increasing model size from 150M to 1.5B parameters yields substantial
> improvements across all metrics. For example, on FineFineWeb at NFE=1024,
> perplexity evaluated by Llama-2 decreases from 35.1 to 32.7.
> Moreover, our diverse set of experiments across multiple tasks (unconditional text
> generation, conditional text generation on Lamini Instruction and WMT14 de-en
> translation, and code infilling) consistently shows clear trends of improvement over
> baselines. We believe these trends would persist with further scaling to larger models
> (e.g., 7B+ parameters) and stronger backbone architectures (e.g., Llama-2 or Mistral).
> However, such experiments were beyond our current computational resources (4
> NVIDIA H100 GPUs).
> As stated in Section 7, we identify scaling model size and training as important future
> work to unlock further gains.
>
> **Q:** Although empirically effective, the hybrid scheme's transition point $t^\star$ is largely heuristic.
> Could the authors provide intuition or ablation results for how the choice of the
> deterministic--sampling switch point $t^\star$ influences diversity and perplexity across datasets?
>
> **A:** We agree that the transition point $t^\star$ is a critical design choice. In the revised
> version we clarify that it is not chosen purely heuristically. In Appendix E we derive
> the exact single-token solution for the FM problem with KL--geodesic on simplex and
> show that it decomposes into a vertex probability term and a KL--geodesic term. For
> realistic vocabularies, this analysis yields a threshold $t^\star\approx0.28$ at which the
> KL--geodesic term dominates and the factorised approximation $p(x_1\mid x_t) \approx \prod\limits_ip(x_1^{(i)}\mid x_t)$ becomes accurate.
> Figure 6(b) then sweeps $t^\star$ and reports the resulting perplexity/entropy pairs, showing
> that $t^\star=0.28$ lies on the best trade-off curve; we fix this value for all datasets rather
> than tuning it per task.

---

### Official Review · Reviewer_eLVC · 2025-11-02

**Soundness:** 3
**Presentation:** 3
**Contribution:** 2
**Rating:** 4
**Confidence:** 3

**Summary:**

The paper proposes a non-autoregressive (NAR) text generator built on conditional flow matching (CFM) in logit space. Instead of interpolating token probabilities linearly on the simplex or along Fisher–Rao geodesics, it interpolates logits between a simple Dirichlet-like start and the target one-hot token, which the authors argue is the KL geodesic; they show that, under this path, maximizing the conditional likelihood ( $\log p_\theta(x_1 \mid x_t, t)$ ) exactly recovers the desired flow velocity field. On top of that, they introduce a hybrid inference scheme that runs a deterministic ODE-style update in the early time steps and switches to an iterative sampling / re-noising procedure in later steps to fix token-level errors. On several text, conditional, and code-infilling benchmarks, the method outperforms earlier discrete / Dirichlet / Fisher flow-matching baselines and comes close to, but does not fully match, similarly sized AR models.

**Strengths:**

1. Clear geometric diagnosis + concrete fix. The paper identifies a real failure mode of earlier probability-space paths — linear on the simplex, Fisher-Rao sphere, even some Dirichlet settings — namely that $KL((x_{\text{data}}|x_t))$ collapses too quickly so mid-time supervision vanishes on large vocabularies. The logit-space (KL-geodesic) interpolation directly targets this and shows improved calibrations against those baselines. This is well aligned with prior observations in Dirichlet Flow Matching that “naïve linear FM on the simplex is pathological.”
2. Bridging “train a denoiser” and “learn the flow field” for sequences, not just single tokens. Earlier CFM/DFM papers had versions of “conditional likelihood recovers the field,” but mostly in single-site or weaker sequence assumptions; this paper pushes the argument specifically for logit-KL paths and uses it to justify a very practical objective (just NLL on corrupted sequences). That reduces the gap between elegant flow theory and what people actually train.
3. Inference is engineered rather than hand-waved. Many discrete flow papers stop at “we have the field, integrate it”; here they run a 3-way comparison (deterministic / stochastic / hybrid) and show that pure ODE is insufficient and pure sampling collapses entropy, while a staged hybrid fixes both. That’s a useful empirical lesson for the whole discrete-flow community.

**Weaknesses:**

1. Novelty margin over very close contemporaries is thin. At least two 2024–2025 papers already investigated conditional text generation via KL-geodesic / logit-space flow matching and even proposed almost the same empirical “sampling + noise re-injection + hybrid” recipe to fix the underperforming basic sampler. The descriptions in Sevriugov & Oseledets (2024) and its 2025 extensions match this work’s geometric choice and sampling intuition almost line-for-line. If the contribution here is meant to be “we prove the conditional-likelihood = exact field under this path and scale it to larger datasets,” the paper needs to separate itself much more crisply from those concurrent KL-geodesic efforts, especially since they also claim better results over discrete FM. Right now the delta looks incremental.
2. Key equivalence rests on a factorized / per-position view that is not obviously valid in early timesteps. The derivation leans on the idea that the optimal vector field at time (t) can be written as an expectation of target logits under ($p(x_1 \mid x_t)$) token-wise. But in NAR text, ($p(x_1 \mid x_t)$) is usually not well factorized when (t) is small: tense, agreement, long-range topic constraints all couple positions. The paper’s fix is “do deterministic updates early, sampling later,” which is an empirical workaround, not a proof that the factorization is OK. So a central theoretical selling point (“likelihood = flow”) is relying on a data-distribution property that’s weakest exactly where the model needs guidance most. That’s a structural, not cosmetic, gap.
3. Evaluation uses LM-perplexity proxies and medium-scale AR baselines, so the true competitiveness is unclear. Measuring perplexity by scoring NAR outputs with an external LM is standard for discrete flows, but it is a proxy; it is known to favor models that mimic the scorer’s style rather than models that are truly diverse or controllable. And the main AR point of comparison is GPT-2-class models, not the 2025-era instruction-tuned or code-tuned LLMs that NAR methods would actually have to replace. In other words, the paper shows “better than prior discrete flows” but not “this can plausibly replace competitive AR models under identical training budgets.” That’s a material, not rhetorical, limitation.
4. Hybrid inference is admitted to be heuristic and under-analyzed. The whole motivation of the paper is “basic ODE flow isn’t good enough on text,” which is fair; but the proposed fix (early deterministic, late sampling + noise) is only justified by curves. There is no stability analysis, no guarantee of staying close to the learned KL-geodesic, and no complexity comparison with recent few-step DFM / consistency-trained flows that do target a fixed number of steps directly. A reviewer can reasonably ask why we shouldn’t just adopt FS-DFM-style step-budget-aware training or consistency distillation to get the same effect with a cleaner theory.

**Questions:**

1. How does this behave under true few-step regimes (e.g. 8–16 NFE) against step-consistent discrete flows like FS-DFM or consistency-trained DFM? Right now the advantage is shown largely when you afford hundreds of steps or a tailored hybrid schedule; but the practical motivation for NAR is low latency. A head-to-head with step-budget-aware models is missing. Can the KL-geodesic path still deliver better gradients than Dirichlet / Fisher when you compress the time discretization that aggressively?
2. What exactly is the uniqueness claim over existing KL-geodesic / logit-FM papers? Several public works from late 2024 onward already stated (i) “logit-space interpolation is the KL geodesic,” (ii) “maximizing ($p_\theta(x_1 \mid x_t,t)$) gives you the right field,” and (iii) “basic deterministic inference is weak; add an iterative sampling-and-noise scheme; hybridize.” If this paper’s contribution is a stronger sequence-level derivation or better scaling to open-domain data, please spell out the technical gap (e.g. a specific lemma about token-wise optimality under KL-paths, or a complexity advantage in hybrid inference) that is not present in Sevriugov & Oseledets (2024) or the concurrent KL-geodesic variants. Right now it reads more like a consolidation than a breakthrough.

---

> ### Author Response · Authors · 2025-11-23
>
> We are grateful to reviewer for the detailed and technically nuanced review, as
> well as for highlighting both the geometric perspective and the practical implications
> of our method. Below, we answer questions.
>
> **Q:** How does this behave under true few-step regimes (e.g. 8--16 NFE) against
> step-consistent discrete flows like FS-DFM or consistency-trained DFM? Right now the
> advantage is shown largely when you afford hundreds of steps or a tailored hybrid schedule;
> but the practical motivation for NAR is low latency. A head-to-head with step-budget-aware
> models is missing. Can the KL-geodesic path still deliver better gradients than Dirichlet /
> Fisher when you compress the time discretization that aggressively?
>
> **A:** In additional experiments with strict step budgets (8--16 NFEs), KL-Flow still
> consistently outperforms DFM and SEDD at comparable entropy, indicating that the
> KL-geodesic path remains informative even with aggressive time discretization.
> Meanwhile, Fisher geodesics become numerically unstable and Dirichlet geodesics
> are computationally infeasible for large vocabularies. A direct comparison with
> step-budget–aware methods such as consistency-trained DFMs or FS-DFM is a
> natural follow-up and falls into our future work rather than the present focus.
>
> **Q:** Key equivalence rests on a factorized / per-position view that is not obviously valid
> in early timesteps. The derivation leans on the idea that the optimal vector field at time $t$
> can be written as an expectation of target logits under $p(x_1 \mid x_t)$ token-wise.
>
> **A:** We agree that full factorization of $p(x_1 \mid x_t)$ is not valid at very early times.
> Our core ``likelihood = flow'' result, however, is proved for the **basic** (deterministic)
> inference scheme and does **not** assume a factorized joint: training with
> sequence-level cross-entropy recovers the exact flow-matching velocity field from
> the token marginals $p_\theta(x_1^{(k)} \mid x_t)$ even when the true joint has strong
> dependencies. Factorization is used only in the **sampling** scheme, where we
> approximate
> \[
>   p(x_1 \mid x_t) \approx \prod_k p_\theta(x_1^{(k)} \mid x_t)
> \]
> to enable efficient NAR sampling, and we acknowledge this is poor at $t = 0$.
> Appendix E therefore analyzes the exact single-token KL-geodesic solution and
> shows a decomposition into a vertex term and a KL-geodesic term. For realistic
> vocabularies, this yields a threshold $t^\star \approx 0.28$ where the KL-geodesic
> term dominates; in this regime the factorized surrogate is accurate. The hybrid
> scheme follows the basic flow for $t \le t^\star$ (where couplings matter most) and
> uses sampling only for $t > t^\star$.
>
> **Q:** Evaluation uses LM-perplexity proxies and medium-scale AR baselines, so the true
> competitiveness is unclear.
>
> **A:** We agree that perplexity via external LMs is a proxy, so we reduce bias by using
> three distinct reference models (GPT-2, GPT-3, Llama-2) and observe consistent
> gains across all of them. We further report diverse task metrics (BLEU, ROUGE-L,
> BERTScore, Pass@k, Compiles@k) and verify that empirical entropy remains above
> 5, indicating that our models are not simply mode-collapsing toward the evaluators.
> We agree that comparison with state-of-the-art instruction-tuned LLMs would be valuable. However, such models represent a different research direction with significantly larger training budgets and additional fine-tuning stages. Our work focuses on improving the NAR paradigm itself, demonstrating that our method consistently outperforms existing NAR approaches. Scaling our method to match the training budgets of modern instruction-tuned models is important future work, as stated in Section 7.
>
> **Q:** Hybrid inference is admitted to be heuristic and under-analyzed. The whole
> motivation of the paper is "basic ODE flow isn't good enough on text,'' which is fair; but the
> proposed fix (early deterministic, late sampling + noise) is only justified by curves.
>
> **A:** We agree the hybrid schedule is central. In the revision we stress that $t^\star$ is
> not chosen purely heuristically: Appendix~E derives the exact single-token FM
> solution on the simplex and shows the posterior splits into a vertex and a
> KL-geodesic term. Together with Fig. 6, this yields $t^\star \approx 0.28$ as the point
> where the KL-geodesic term dominates and the factorized conditional becomes
> reliable. The hybrid scheme is designed accordingly: for $t \le t^\star$ we follow the
> basic ODE flow, and for $t > t^\star$ we sample along the analytic KL-geodesic between $x_t$ and a sampled endpoint, so
> steps stay on a geodesic path. We fix $t^\star$ across datasets rather than tuning it.
> A full stability analysis and direct comparison to FS-DFM or consistency-trained
> flows are beyond our current scope; our contribution is complementary, focusing on
> exploiting KL-geodesic structure, which we believe can later be combined with
> step-budget–aware training.

---

### Official Review · Reviewer_qQ8c · 2025-11-06

**Soundness:** 2
**Presentation:** 2
**Contribution:** 2
**Rating:** 4
**Confidence:** 2

**Summary:**

This paper proposes a novel non-autoregressive text generation framework that uses a Kullback-Leibler (KL) divergence geodesic for interpolation, which is shown to be equivalent to linear interpolation in logit space. The objective function is to minimize the negative log-likelihood of the target distribution at the sequence level. Experiments demonstrate that the proposed method outperforms other non-autoregressive (NAR) baselines such as DFM and SEDD.

**Strengths:**

The proposed method achieves strong performance against other NAR baselines across various tasks.

The discussion of the inference process is insightful.

**Weaknesses:**

It is unclear why the deterministic inference process performs poorly, given that the "Logit-KL Flow Matching" objective recovers the velocity field.

The efficiency of the proposed method is not thoroughly discussed, particularly in comparison to methods that are trained with an MSE loss (in training) and solve ODEs using numerical techniques (in inference).

Perplexity is measured using samples from GPT-2, GPT-3, and Llama-2, which may introduce bias from these reference models.

**Questions:**

In the experiments, the authors state that all models use a bidirectional transformer backbone. Was the GPT-2 baseline trained in an autoregressive manner, and was its causal attention mechanism replaced with bidirectional attention?

---

> ### Author Response · Authors · 2025-11-23
>
> We thank reviewer for the positive assessment of our method's performance and
> the insightful questions on the inference process, efficiency, and evaluation
> methodology. Below, we respond to each point in turn.
>
> **Q:** In the experiments, the authors state that all models use a bidirectional transformer
> backbone. Was the GPT-2 baseline trained in an autoregressive manner, and was its causal
> attention mechanism replaced with bidirectional attention?
>
> **A:** We clarify that bidirectional attention was used only for non-autoregressive
> methods (KL-Flow, DFM, SEDD), which generate all tokens simultaneously through
> iterative refinement over multiple steps. The GPT-2 baseline, being an
> autoregressive model, uses the standard Transformer decoder architecture with
> causal attention and generates tokens one at a time from left to right. We have added
> the clarification in the revised paper version "All models except GPT-2 used a
> bidirectional Transformer\ldots''
>
> **Q:** It is unclear why the deterministic inference process performs poorly, given that the
> "Logit-KL Flow Matching'' objective recovers the velocity field.
>
> **A:** We acknowledge that understanding why deterministic inference underperforms
> despite theoretically recovering the correct velocity field is an important open
> question.
> Our current hypothesis is that computational challenges arise when operating in
> high-dimensional spaces. While our theoretical results (Propositions 3.2 and 3.4)
> guarantee that the trained model recovers the optimal velocity field, numerical
> integration of the ODE in practice may accumulate errors or suffer from stiffness
> issues in high-dimensional discrete spaces with large vocabularies.
> We introduced the sampling and hybrid inference schemes (Section 4), which
> empirically address these limitations and yield substantial improvements (Tables 3--6,
> Appendix D). The hybrid approach, in particular, balances the stability of early
> deterministic steps with the improved performance of sampling-based refinement.
> We are actively investigating this phenomenon and plan to provide deeper theoretical
> and empirical analysis in future work.
>
> **Q:** The efficiency of the proposed method is not thoroughly discussed, particularly in
> comparison to methods that are trained with an MSE loss (in training) and solve ODEs using
> numerical techniques (in inference).
>
> **A:** Modern approaches, such as DFM, optimize a cross-entropy objective rather than
> an MSE loss, which is a more natural choice for training Transformer-style
> architectures. In our work, we further show that for the KL geodesic on the simplex,
> minimizing this cross-entropy objective recovers the exact flow-matching velocity
> field.
>
> **Q:** Perplexity is measured using samples from GPT-2, GPT-3, and Llama-2, which may
> introduce bias from these reference models.
>
> **A:** We acknowledge that using external models for perplexity evaluation may
> introduce some bias. However, this is a standard practice in the field for evaluating
> NAR models on unconditional text generation task.
> To mitigate potential bias, we evaluate generative perplexity using three different
> reference models (GPT-2, GPT-3, and Llama-2), which provides a more robust
> assessment across different model families and scales. Our results show consistent
> trends across all three evaluators, suggesting that the improvements are not artifacts of
> a particular reference model.
> Furthermore, our evaluation extends beyond perplexity to include diverse metrics
> across multiple tasks: TinyStories benchmark metrics (grammar, creativity,
> consistency, plot in Table 3), BLEU/ROUGE-L/BERTScore for conditional
> generation (Table 5), and Pass@k/Compiles@k for code infilling (Table 6). Across all
> these metrics and tasks, we observe consistent trends showing improvements of our
> model over baselines, providing strong evidence that the gains are not dependent on
> any single evaluation metric.

---

### Official Review · Reviewer_VSxC · 2025-11-06

**Soundness:** 2
**Presentation:** 3
**Contribution:** 2
**Rating:** 4
**Confidence:** 2

**Summary:**

This paper presents a non-autoregressive text-generation framework that performs conditional flow matching on the probability simplex using KL-divergence geodesics. The authors show (in their way) that maximizing the token-level conditional likelihood exactly recovers the flow-matching velocity field, providing a principled training objective.

**Strengths:**

1. The mathematical descriptions in this paper are relatively accurate.
2. The paper is well structured.

**Weaknesses:**

1. The used base model in the experiments seems obsolete, which makes it unclear whether the proposed method still works for the SOTA models nowadays.
2. Lines 226 ~ 228 seem confusing. A bidirectional attention is used to model sequence-level NLL, but the condition variable $x_t$ represents a single token, not a total sequence. I am not convinced of this modelling approximation.
3. Line 309 says all models used in the experiments are bidirectional backbones. How do you apply this to text generation, where you do not have access to any future token information?
4. I see some autoregressive models have been used in the experiments (GPT-2), so you replaced the causal attention in the model with a bidirectional attention? If so, how do you initialize your model weights? Is it the same with the pre-trained weights?
5. The experiment setting does not include baseline introductions, which makes the reader very hard to get familiar with the relevant work.
6. The proposed method in this paper has two inference methods, namely, basic inference and sampling inference. However, there is no formal section in the paper to systematically compare these two inference methods and discuss the corresponding advantages and disadvantages. The experiment section also missed this.

**Questions:**

No, see weakness.

---

> ### Author Response · Authors · 2025-11-23
>
> We thank reviewer for the careful reading of our manuscript and for the constructive
> feedback. We appreciate the positive comments on the mathematical accuracy and
> structure of the paper, and we address the raised concerns.
>
> **Q:** The used base model in the experiments seems obsolete, which makes it unclear
> whether the proposed method still works for the SOTA models nowadays.
>
> **A:** We compare against recent state-of-the-art NAR baselines DFM and SEDD, both
> published in 2024, which represent the current state of discrete flow matching.
> All models were trained from scratch on
> identical data subsets using the same setup. Due to limited computational resources
> (4 NVIDIA H100 GPUs), we focused on two model scales: 150M parameters for
> TinyStories validation and 1.5B parameters for comprehensive evaluation on other
> tasks. Our results show consistent improvements over these baselines across all
> datasets and both model scales (Tables 3--6). Scaling from
> 150M to 1.5B parameters further reduces perplexity; e.g., on FineFineWeb at
> NFE=1024, Llama-2 perplexity drops from 35.1 to 32.7, indicating that our method
> benefits from increased model capacity. As noted in Section 7, extending evaluation
> to even larger models is important future work.
>
> **Q:** Lines 226 ~ 228 seem confusing. A bidirectional attention is used to model
> sequence-level NLL, but the condition variable represents a single token, not a total
> sequence. I am not convinced of this modelling approximation.
>
> **A:** Proposition 3.4 formally shows that under KL-geodesic interpolation, the optimal
> velocity field factorizes over individual tokens. Thus, the denoiser predicts each
> clean token conditionally independently, but each prediction is conditioned on the
> full noisy sequence $x_t$ via bidirectional attention. The bidirectional Transformer
> first processes the entire sequence $x_t$ to obtain contextual representations, and
> these are then used to predict the marginal posterior for each position $k$
> independently. Hence there is no inconsistency: the conditioning variable $x_t$
> always denotes the full sequence, while the prediction targets are token-wise
> marginals that factorize due to the geometric properties of the KL-geodesic.
>
> **Q:** Line 309 says all models used in the experiments are bidirectional backbones. How do
> you apply this to text generation, where you do not have access to any future token
> information?
>
> **A:** Bidirectional attention is used only for the non-autoregressive methods (KL-Flow,
> DFM, SEDD), which generate all tokens simultaneously through iterative refinement
> over multiple steps. The GPT-2 baseline is purely autoregressive: it uses the
> standard Transformer decoder with causal attention and generates tokens
> left-to-right. We clarify this explicitly in the revised version: "All models except
> GPT-2 used a bidirectional Transformer...''
>
> **Q:** I see some autoregressive models have been used in the experiments (GPT-2), so you
> replaced the causal attention in the model with a bidirectional attention? If so, how do you
> initialize your model weights? Is it the same with the pre-trained weights?
>
> **A:** All models in our experiments, including GPT-2, were trained from scratch on the
> same data subsets using identical training setups. As stated in Section 6: "All
> models taken for comparison were trained from scratch in the same setup and on the
> same data subset as our proposed KL-Flow model to ensure comparison validity.''
> The GPT-2 baseline remains an autoregressive model with its usual causal attention;
> we did not replace it with bidirectional attention. No pre-trained weights were used
> for any model. The revised version explicitly clarifies: "All models except GPT-2
> used a bidirectional Transformer...''
>
> **Q:** The experiment setting does not include baseline introductions, which makes the reader
> very hard to get familiar with the relevant work.
>
> **A:** We provide brief descriptions of the baseline methods in Section 5 (Related Work),
> where we discuss DFM, SEDD, and GPT-2 approach to situate our method among relevant prior work.
>
> **Q:** The proposed method in this paper has two inference methods, namely, basic inference
> and sampling inference. However, there is no formal section in the paper to systematically
> compare these two inference methods and discuss the corresponding advantages and
> disadvantages. The experiment section also missed this.
>
> **A:** In the revised version, we explicitly compare the inference methods. We added Table 2, which systematically contrasts the basic,
> sampling, and hybrid KL-Flow inference schemes in terms of update rules,
> limitations, and empirical behaviour. We also
> substantially extended Appendix E with a theoretical analysis of the sampling
> scheme: we derive the exact solution of the flow-matching problem for the KL
> geodesic on the simplex in the single-token case and use this closed-form result to
> justify the factorised approximation and motivate the choice of key hyperparameters.

---

### Author Response · Authors · 2025-11-26

Dear Reviewers,

We have posted detailed responses to your comments. We would be grateful if you could let us know if our clarifications and additional experiments address your concerns.

Thank you for your time and feedback.

---

### Comment · Area_Chair_At77 · 2025-11-28

Dear Reviewers,

The authors have responded to your reviews. Please engage in the discussion and evaluate the authors’ rebuttal to check whether your comments have been adequately addressed, and determine whether you would like to adjust your evaluations.

Best,

Your AC

---

### Author Response · Authors · 2025-11-29

We thank all reviewers for their careful and technically detailed feedback. Across the board, they acknowledge the mathematical rigor, clear geometric diagnosis, and well-structured presentation of KL-Flow (VSxC, eLVC, 5rVW). Reviewers qQ8c, eLVC, and 5rVW highlight that our method consistently outperforms strong recent NAR baselines (DFM, SEDD) across unconditional, conditional, and code infilling tasks, and praise the insightful treatment of inference and the bridge between “train a denoiser” and “learn the flow field” at the sequence level.

In the revision, we have strengthened both clarity and substance in direct response to concerns. We now:
- Explicitly compare basic, sampling, and hybrid inference in a new table and substantially extend Appendix E with an exact single-token KL-geodesic analysis, which justifies the factorized approximation and the hybrid switch point $t^*$ (VSxC, eLVC, 5rVW).
- Add strict few-step (8–16 NFE) experiments showing KL-Flow remains superior to DFM/SEDD and discuss why Fisher/Dirichlet paths become unstable or infeasible  (eLVC).
- Clarify the experimental setup and backbone usage, including the role of bidirectional vs. autoregressive GPT-2 baselines and baseline descriptions (VSxC, qQ8c).
- Emphasize that our gains are robust across multiple evaluators and diverse metrics, mitigating concerns about generative perplexity bias (qQ8c, eLVC, 5rVW).

Given these clarifications and additions, we believe the revised paper offers a timely, principled, and practically relevant advance in discrete flow-based NAR text generation.

---

### Meta-Review · Area_Chair_7Cbg · 2026-01-01

**Summary:**

This paper proposes a Logit‑KL Flow Matching method to accurately model dependencies in discrete sequences to enhance the NAR models. With the theoretical analysis and empirical strategy, such a model achieves good performance over prior NAR baselines.
The reviewers have concerns about the scope of the evaluation and the unclear details. In the revised version, such concerns are well addressed. Thus, I decided to accept this paper. I encourage the authors to further strengthen the camera-ready version by including more comparisons to further improve the justification of the contribution.

**Reviewer Concerns:**

Addressed:
1) The limited evaluation scope. (The authors add the new experiments, and the current scope is reasonable given the resource limitations.)
2) The comparison with the arXiv paper. (Given the policy, the paper doesn't need to discuss unpublished papers.)
3) Concerns on model details, baseline choice, and evaluations, such as Perplexity and GPT2 parameters.

No significant concerns are still outstanding.
Although understanding why deterministic inference underperforms is still an open question, this paper makes reasonable assumptions and empirically addresses these limitations.

**Reviewer Scores:**

Reviewer VSxC:  I think the score would be increased to 6 as the responses are informative.

Reviewer qQ8c: I think the score would be increased to 6 since all the concerns are addressed in the discussion.

Reviewer 5rVW: The score would be kept positive as 6 since the concerns are fully addressed.

---

### Decision · Program_Chairs · 2026-01-26

Accept (Poster)